



# Experimental investigation into the volatilities of highly oxygenated organic molecules (HOM)

Otso Peräkylä[1], Matthieu Riva[1,2], Liine Heikkinen[1], Lauriane Quéléver[1], Pontus Roldin[3], and
Mikael Ehn[1]

[1]Institute for Atmospheric and Earth System Research / Physics, Faculty of Science, University of Helsinki, Finland
[2]Univ Lyon, Université Claude Bernard Lyon 1, CNRS, IRCELYON, F-69626, Villeurbanne, France
[3]Division of Nuclear Physics, Lund University, P.O. Box 118, 22100 Lund, Sweden

**Correspondence:** Otso Peräkylä (otso.perakyla@helsinki.fi) and Mikael Ehn (mikael.ehn@helsinki.fi)

**Abstract.** Secondary organic aerosol (SOA) forms a major part of the tropospheric submicron aerosol. Still, the exact formation mechanisms of SOA have remained elusive. Recently, a newly discovered group of oxidation products of volatile organic compounds (VOC), highly oxygenated organic molecules (HOM), have been proposed to be responsible for a large fraction of SOA formation. To assess the potential of HOM to form SOA, and to even take part in new particle formation, knowledge

of their exact volatilities would be essential. However, due to their exotic, and partially unknown, structures, estimating their volatility is challenging. In this study, we performed a set of continuous flow chamber experiments, supported by box modelling, to study the volatilities of HOM, along with some less oxygenated compounds, formed in the ozonolysis of $\alpha$-pinene, an abundant VOC emitted by the boreal forests. Along with gaseous precursors, we periodically injected inorganic seed aerosol into the chamber to vary the condensation sink (CS) of low volatility vapours. We monitored the decrease of oxidation products

in the gas phase in response to increasing CS, and were able to relate the responses to the volatilities of the compounds. We found that HOM monomers are mainly of low volatility, with a small fraction being semi-volatile. HOM dimers were all of at least low volatility, but probably of extremely low volatility: however, our method is not directly able to distinguish between the two. We were able to explain the volatility of the oxidation products in terms of their carbon, hydrogen, oxygen and nitrogen numbers. We found that increasing levels of oxygenation correspond to lower volatilities, as expected, but that the decrease is

less steep than would be expected based on many existing models for volatility, such as SIMPOL. The hydrogen number of a compound also explained its volatility, independently of the carbon number, with higher hydrogen numbers corresponding to lower volatilities. This can be explained in terms of the functional groups making up a molecule: high hydrogen numbers are associated with e.g. hyrdoxy groups, which lower the volatility more than e.g. carbonyls, that are associated with a lower hydrogen number. The method presented should be applicable to systems other than $\alpha$-pinene ozonolysis, and with different

organic loadings in order to study different volatility ranges.

## 1   Introduction

Secondary organic aerosol (SOA) makes up a major fraction of the tropospheric submicron aerosol world-wide (Zhang et al., 2007; Jimenez et al., 2009). Despite its importance and much research effort, our fundamental understanding of the formation



of SOA has remained lacking (Hallquist et al., 2009; Shrivastava et al., 2017). Recently, a new group of oxidation products of volatile organic compounds (VOCs), highly oxygenated organic molecules (HOM, Ehn et al., 2014, 2017; Bianchi et al., 2019), has been proposed to be the source of a major fraction of tropospheric submicron SOA (Ehn et al., 2014). Important sources of HOM include many monoterpenes, the most abundant group of biogenic VOCs emitted by boreal forests.

Most VOCs form organic peroxy ($RO_2$) radicals upon oxidation. HOM form through the autoxidation of these organic peroxy radicals, a process only recently discovered to be important in atmospheric oxidation (Crounse et al., 2013; Ehn et al., 2014). An extensive description of HOM formation can be found in Bianchi et al. (2019). Briefly, in autoxidation, $RO_2$ radicals undergo intramolecular hydrogen abstractions, followed by the addition of molecular oxygen on the resulting carbon-centered radical. This results in a new $RO_2$ radical, with an additional hydroperoxy group. This process can repeat multiple times. The

radical reaction chain can be terminated unimolecularly, through the loss of an OH radical from the $RO_2$ radical, resulting in a closed shell oxidation product. It can also be terminated bimolecularly, for example through the reaction with another $RO_2$ radical, or nitric oxide (NO). The termination mechanism is important in determining which types of HOM form in the reaction. As an example, the bimolecular termination with another $RO_2$ radical can very efficiently lead to the formation of molecular dimers of extremely low volatility (Berndt et al., 2018b, a), and the termination with NO can lead to the formation

of organic nitrates. Due to the fast autoxidation process, combined with the different termination mechanisms, the products can rapidly acquire high oxygen contents, with those having at least six oxygen atoms counted as HOM (Bianchi et al., 2019). The oxygen appears in the form of many functional groups, including carbonyl, hydroperoxy, hydroxy, peroxy acid and carboxylic acid groups (e.g. Mentel et al., 2015).

Due to the high level of functionalization, HOM are thought to be of low volatility (Bianchi et al., 2019). This is supported

by observations that HOM are efficient in forming SOA, and even able to take part in the very first steps of new particle formation (NPF) (Ehn et al., 2014; Kirkby et al., 2016). To determine exactly to what extent HOM can impact NPF and SOA formation, the knowledge of their vapour pressures is essential (Bianchi et al., 2019). Still, determining the vapour pressures of HOM remains challenging. Most of the species have not been isolated, hampering experimental characterization (Bianchi et al., 2019). Due to the high numbers of functional groups, estimating the vapour pressures with commonly used functional group

contribution methods is not sufficient, and different computational approaches give estimates of the vapour pressures spanning many orders of magnitude (Kurtén et al., 2016). In order to investigate the volatility of HOM formed in the ozonolysis of $\alpha$-pinene, Ehn et al. (2014) used injections of inorganic seed aerosol to increase the condensation sink in a chamber experiment. They observed the behaviour of HOM to be consistent with kinetically limited condensation and extremely low volatilities, but only looked at the sum of HOM, not focusing on individual compounds. Thus, HOM were initially classified as extremely

low volatility organic compounds (ELVOC, according to the definitions by Donahue et al., 2012). Theoretical work by Kurtén et al. (2016) has later suggested that the most HOM, especially the monomers, might not be ELVOC, but low volatility organic compounds or even semi-volatile organic compounds (LVOC and SVOC respectively, again according to Donahue et al., 2012). Thus, detailed information on HOM volatility is still lacking, which presents a key obstacle for assessing their impact on particle formation (Bianchi et al., 2019).





Here, we investigate the volatility of HOM, along with some less oxygenated compounds, experimentally, in a manner similar to Ehn et al. (2014), but on a molecular level. We produce HOM in the ozonolysis of $\alpha$-pinene in a continuous flow smog chamber, and measure them in the gas phase with the nitrate Chemical Ionization Atmospheric Pressure interface Time of Flight mass spectrometer (CI-APi-TOF, Jokinen et al., 2012). We inject inorganic seed aerosol into the chamber, and monitor

how the gas phase concentrations of the oxidation products change. We show that we can gain valuable and detailed information about the volatility of the oxidation products using this method. Further, we develop a parametrization describing the volatility of HOM formed in the ozonolysis of $\alpha$-pinene, especially the monomers, in terms of their composition, and compare the results to existing parametrizations and estimates on HOM volatility.

## 2  Methods

### 2.1  Chamber set-up

In order to investigate the volatility of HOM formed in the gas phase in a controlled manner, we conducted a series of laboratory experiments in the newly constructed COALA chamber at the University of Helsinki, previously presented by Riva et al. (2019). The chamber is a two cubic metre bag, made of teflon (FEP) foil with dimensions (L × W × H) 1650 × 1100 × 1100 mm and 0.125 mm thickness, supplied by Vector Foiltec (Bremen, Germany). The chamber is contained in a rigid enclosure with 400

15  nm LED lights to photolyse nitrogen dioxide ($NO_2$) to NO, and is stirred with a teflon fan to ensure homogeneous mixing of the air inside. The chamber was operated in a continuous flow mode, with a residence time of 50 minutes, attained by setting the inflow to the chamber to 40 litres per minute (Lpm). In the continuous flow mode the total flow out of the chamber was the same as the inflow. The instrumentation (Sect. 2.2) sampled the majority of the flow out of the chamber. The chamber was maintained at a slight overpressure, resulting in the rest of the flow being flushed to an exhaust line.

We injected dry air purified with a clean air generator (AADCO model 737-14, Ohio, USA) into the chamber, along with gaseous reactants $\alpha$-pinene, $O_3$, and in some of the experiments, $NO_2$. Ozone was generated by injecting purified air through an ozone generator (Dasibi 1008-PC), while $\alpha$-pinene and $NO_2$ were from gas bottles. The injections were controlled with a range of mass flow controllers (MKS, G-Series, 0.05-50 Lpm, Andover, MA, USA). We controlled the relative humidity in the chamber to be either < 1 % or 40 %: this was done by bubbling the dry clean air through a bubbler filled with purified (Milli-Q)

water.

In addition to the gaseous reactants, we injected size selected, 80 nm inorganic seed particles, consisting of either ammonium sulfate (AS) or ammonium bisulfate (ABS), into the chamber. ABS was used in order to promote acidity-dependent particle phase reactions: the effect of these on the particle phase has been presented by Riva et al. (2019). The particles were produced by atomizing a solution consisting of Milli-Q water and AS or Milli-Q water and ABS, after which the seed particles were

dried for size selection. After size selection, the particles were either injected into the chamber as dry, or subjected to a relative humidity of over 80 % to attain deliquescence before injection into the chamber. In the < 1 % RH experiments, we only used the dried particles, while in the 40 % RH experiments, both types were used.



## 2.2 Instrumentation

We monitored the chamber with a suite of online instruments measuring both gas and particle phases. For measuring the responses of HOM and other oxidation products of $\alpha$-pinene to the seed injections, we used the nitrate CI-APi-TOF (TOFWERK AG, Aerodyne, Jokinen et al., 2012), equipped with a long time-of-flight (LTOF) mass analyzer for a resolving power of over 13 000 in the HOM mass range (300 – 650 Da). The instrument can detect highly oxygenated compounds as clusters with the nitrate ion ($NO_3^-$). It allows for the determination of the molecular formulae of the detected clusters due to its high mass resolution. Along with closed shell products, the CI-APi-TOF can detect certain highly oxidized $RO_2$ radicals as well. As the method is based on the soft clustering of the analyte molecules with the nitrate anion, we assume that any multiple charging effects are negligible. Thus, the charge that the detected ions carry is always assumed to be -$e$, and the measured mass-to-charge ratios to directly correspond to masses. We express the masses in daltons (Da).

We used a Proton Transfer Reaction Time-Of-Flight mass spectrometer (PTR-TOF 8000, Ionicon, Graus et al., 2010) to measure the concentration of the reactant $\alpha$-pinene, and a UV photometric ozone monitor (Model 49p, Thermo Environmental Instruments) for the ozone concentration. For measuring the $NO_X$ concentration we used a chemiluminescence $NO-NO_2-NO_X$ analyzer (Model 42i, Thermo Fisher Scientific), and for the NO concentration the more sensitive Ecophysics CLD 780 TR.

For the particle phase measurements, we used a differential mobility particle sizer (DMPS, Aalto et al., 2001) and a high resolution aerosol mass spectrometer (HR-AMS, DeCarlo et al., 2006), also equipped with the LTOF analyzer. We used the DMPS for the determination of the dry aerosol size distribution in the chamber between 10 and 400 nm, and the AMS for the chemical composition of the dried particles.

From the DMPS size distribution, we also calculated the dry condensation sink (CS), describing the ability of a particle size distribution to remove low-volatility vapours from the gas phase (Dal Maso et al., 2005). CS depends on the molecular diffusion coefficient and mean molecular speed of a compound: thus, the value of CS is compound-dependent. Instead of the often used values for sulfuric acid, we calculated the condensation sink specifically for HOM. The molecular diffusion coefficient was calculated using Fuller's method (Tang et al., 2015), and the mean molecular speed was calculated using kinetic theory. Both the molecular diffusion coefficient and speed depend on molecular composition and on the absolute temperature during the experiments. The values presented for the condensation sink are calculated for the compound $C_{10}H_{16}O_7$, and are around 40 % lower than for sulfuric acid. In comparison, the CS calculated for the largest molecules (i.e. HOM dimers) were approximately 30 % lower than for $C_{10}H_{16}O_7$.

## 2.3 Overview of experiments

In a typical experiment, we first continuously injected only gaseous precursors $\alpha$-pinene, $O_3$, and in some of the experiments, $NO_2$, into the dry or humidified chamber. In the experiments with $NO_2$, we also used 400 nm LED lights to photolyse $NO_2$ to NO. We used two intensities for the LED lights: these corresponded to steady state NO concentrations of around 100 and 200 ppt with around 30 ppb of $NO_2$. During the injection of gaseous precursors, we observed particle formation. The injection was continued until a steady state had been reached with respect to both the gas phase and the particle phase. After sampling the





steady state chamber for a number of hours, we started injecting either AS or ABS particles, either dried or deliquesced. The injection was continued until a new steady state had been reached, and been sampled for a number of hours. The duration of a typical experiment was eight hours without the inorganic seed, and eight hours of seed injection. The temperature during the experiments was around 302 K. An overview of experimental condition is presented in Table A1.

**2.4   Continuous flow chamber dynamics**

The time evolution of the gas phase concentration of a compound in the chamber is determined by its sinks ($S$) and sources ($Q$). The sources of a compound to the gas phase consist of its injection into the chamber, its chemical production in the gas phase in the chamber, and its evaporation from chamber walls and aerosol particles. Its sinks, on the other hand, consist of its flush out from the chamber, its loss to chemical reactions, and its condensation onto walls and aerosol particles. In an actively

mixed chamber, such as the one used, the concentration of compounds is homogeneous across the chamber. Thus, the effect of sources and sinks on the concentration of a compound X in the chamber can be expressed as follows:

$$\frac{d[X]}{dt} = Q_{\text{injection}} + Q_{\text{chemical}} + Q_{\text{wall}} + Q_{\text{aerosol}} - S_{\text{flush out}} - S_{\text{chemical}} - S_{\text{walls}} - S_{\text{aerosol}}. \tag{1}$$

In a continuous flow chamber, given that the inflow of reactants is kept constant, a steady state is eventually reached. In a steady state, the sources and sinks of a compound are equal, and thus its time derivative in Eq. (1) goes to zero and its

concentration stays constant. The time required for the formation of a steady state varies between components in the chamber. In the following section we present some limiting cases of gas phase compounds and the steady states formed between their sources and sinks.

**2.4.1   Effect of volatility on the behaviour of compounds in the gas phase**

The volatility of a gas phase compound affects the type of steady state it forms in the chamber. Next we will qualitatively

outline which terms in Eq. (1) are important for different types of compounds, and how those affect the types of steady states, as well as the sensitivities of those steady states to seed injections to the chamber.

For volatile reactants like $\alpha$-pinene (AP in equations), the loss by condensation to either chamber walls or aerosol particles is negligible. In other words, condensation will be followed by prompt evaporation back to the gas phase. In this case, the condensation and evaporation terms in Eq. (1) can be omitted. Furthermore, $\alpha$-pinene is not chemically produced in the gas

phase, but injected into the chamber. Thus, the injection is the only source term remaining, while the loss terms are the chemical loss and flush out of the chamber. In a steady state, we can write the Eq. (1) for $\alpha$-pinene as follows:

$$Q_{\text{AP, injection}} = S_{\text{AP, flush out}} + S_{\text{AP, chemical}}. \tag{2}$$

Injection rate of $\alpha$-pinene is kept constant, independently of the concentration in the chamber. In contrast, the rate at which $\alpha$-pinene is flushed out of the chamber is directly proportional to its concentration in the air leaving the chamber, which is the





same as the concentration in the chamber. The chemical sink is caused by the oxidation of $\alpha$-pinene, and is dependent on the concentrations of both $\alpha$-pinene and its oxidants. In this study, we injected ozone into the chamber to oxidize $\alpha$-pinene. In the ozonolysis reactions of alkenes, hydroxyl (OH) radical is also produced: for $\alpha$-pinene, the yield is close to one (Atkinson et al., 1992; Paulson and Orlando, 1996). OH goes on to react with $\alpha$-pinene. Further, in the experiments with $NO_2$ injected, some
nitrate ($NO_3$) radical is produced, which also reacts with $\alpha$-pinene. We can thus expand Eq. (2):

$$Q_{\text{AP, injection}} = S_{\text{AP, flush out}} + S_{\text{AP, chemical}}$$
$$= (k_{\text{flush out}} + k_{O_3+\text{AP}}[O_3]_{\text{SS}} + k_{\text{OH}+\text{AP}}[\text{OH}]_{\text{SS}} + k_{NO_3+\text{AP}}[NO_3]_{\text{SS}})[\text{AP}]_{\text{SS}}$$
$$[\text{AP}]_{\text{SS}} = \frac{Q_{\text{AP, injection}}}{k_{\text{flush out}} + k_{O_3+\text{AP}}[O_3]_{\text{SS}} + k_{\text{OH}+\text{AP}}[\text{OH}]_{\text{SS}} + k_{NO_3+\text{AP}}[NO_3]_{\text{SS}}}, \qquad (3)$$

where SS refers to steady state conditions, and $k_{\text{flush out}}$ is the reciprocal of the chamber turnover time, 50 minutes. $Q_{\text{AP}}$
represents the injection rate of $\alpha$-pinene into the chamber, which is kept constant throughout an experiment. Thus, the steady state $\alpha$-pinene concentration is determined by its input to the chamber, the steady state concentrations of the oxidants, and the turnover time.

Similarly to $\alpha$-pinene, oxidation products of relatively high volatility, such as intermediate volatility organic compounds (IVOC, Donahue et al., 2012), do not readily condense on aerosol particles with loadings used in our experiment (Table A1).
However, it is possible that some of the compounds partition onto the chamber walls. This is due to the observation that, with respect to the partitioning of organic vapours, Teflon walls of smog chambers behave in a manner similar to a very high organic aerosol loading (e.g. Matsunaga and Ziemann, 2010; Krechmer et al., 2016). The extent of the partitioning of the compounds to the walls varies depending on e.g. their volatility. Thus, the evaporation and condensation terms related to aerosol particles in Eq. (1) can be assumed negligible, but the corresponding terms for wall interactions not.
The oxidation products of $\alpha$-pinene generally have lost their carbon-carbon double bond upon the initial oxidation reaction. This means that they are unreactive towards ozone, but can still be oxidized by the OH and $NO_3$ radicals formed in the chamber. Thus, the sinks of gas phase IVOC in the chamber include their potentially reversible loss to chamber walls, flush out of the chamber, and chemical sink to reactions with radicals. As the oxidation products are not directly injected into the chamber, but produced in the oxidation of $\alpha$-pinene, the sources include their chemical production and evaporation from walls. Therefore,
we can express their steady state concentration as follows:

$$Q_{\text{IVOC, chemical}} + Q_{\text{IVOC, wall}} = S_{\text{IVOC, flush out}} + S_{\text{IVOC, chemical}} + S_{\text{IVOC, walls}}$$
$$= (k_{\text{flush out}} + k_{\text{OH}+\text{IVOC}}[\text{OH}]_{\text{SS}} + k_{NO_3+\text{IVOC}}[NO_3]_{\text{SS}} + k_{\text{IVOC, walls}})[\text{IVOC}]_{\text{SS}}$$
$$[\text{IVOC}]_{\text{SS}} = \frac{Q_{\text{IVOC, chemical}} + Q_{\text{IVOC, wall}}}{k_{\text{flush out}} + k_{\text{OH}+\text{IVOC}}[\text{OH}]_{\text{SS}} + k_{NO_3+\text{IVOC}}[NO_3]_{\text{SS}} + k_{\text{IVOC, walls}}}, \qquad (4)$$

where the $k_{\text{IVOC, walls}}$ is the wall loss rate coefficient for IVOC, and evaporation from walls is expressed in terms of the wall
source. Here the sink of the IVOC is thus independent of the condensational aerosol surface area in the chamber. This means





that the volatility of the compound is high enough that there is no net condensation in the time scale of the chamber turnover. It is important to note that the wall source for IVOC is not necessarily constant, but may depend on temperature, humidity and chamber history through the accumulation of IVOC on the chamber walls.

Like IVOC, oxidation products of low volatility, such as ELVOC, are not directly injected into the chamber, but produced

5   from $\alpha$-pinene oxidation in the gas phase. Because of their extremely low volatility, their evaporation into the gas phase from either aerosol particles or chamber walls is negligible. Instead, they are lost by irreversible condensation to the particles and walls. In addition, they are flushed out of the chamber, and lost by oxidation reactions with radicals. In a steady state, the sources and sinks are equal and we can write

$$Q_{\text{ELVOC, chemical}} = S_{\text{ELVOC, flush out}} + S_{\text{ELVOC, chemical}} + S_{\text{ELVOC, walls}} + S_{\text{ELVOC, aerosol}}$$

$$= (k_{\text{flush out}} + k_{\text{OH+ELVOC}}[\text{OH}]_{\text{SS}} + k_{\text{NO}_3+\text{ELVOC}}[\text{NO}_3]_{\text{SS}} + k_{\text{ELVOC, walls}} + \text{CS}_{\text{ELVOC}})[\text{ELVOC}]_{\text{SS}}$$

$$[\text{ELVOC}]_{\text{SS}} = \frac{Q_{\text{ELVOC, chemical}}}{k_{\text{flush out}} + k_{\text{OH+ELVOC}}[\text{OH}]_{\text{SS}} + k_{\text{NO}_3+\text{ELVOC}}[\text{NO}_3]_{\text{SS}} + k_{\text{ELVOC, walls}} + \text{CS}_{\text{ELVOC}}}, \qquad (5)$$

where $\text{CS}_{\text{ELVOC}}$ is the condensation sink for ELVOC, caused by aerosol particles, and $k_{\text{ELVOC, walls}}$ is the wall loss rate. The sinks together determine the average lifetime of an ELVOC molecule in the gas phase, and Eq. (5) can also be written in terms of the lifetime, $\tau_{\text{ELVOC}}$:

$$[\text{ELVOC}]_{\text{SS}} = Q_{\text{ELVOC, chemical}} \tau_{\text{ELVOC}}. \qquad (6)$$

We estimate, from their behaviour upon seed addition, that the wall loss lifetime of ELVOC is on the order of 400 seconds in the COALA chamber. The flush out lifetime is the same as the chamber residence time, 50 minutes, or 3000 seconds. We do not know the exact reaction rate coefficients between ELVOC species and OH, or the OH concentration in the chamber. To get an upper limit for the chemical loss, we can assume a collision limited reaction similarly to Bianchi et al. (2019), and

production of OH from every ozone-$\alpha$-pinene reaction, with $\alpha$-pinene acting as the main OH sink. With these assumptions, we estimate the lifetime of ELVOC towards OH radical reactions to be on the order of 2000 seconds. Using similar reasoning, the contribution of reactions with the nitrate radical should be at maximum comparable to the OH reactions, but probably much smaller. This means that without any aerosol in the chamber, the majority of ELVOC are lost to condensation onto chamber walls (Fig. 1). A typical condensation sink caused by particles formed in the chamber in the absence of inorganic

seed was $2 \times 10^{-3}$ s$^{-1}$ (Table A1), corresponding to a lifetime of 500 seconds with respect to the loss to particle surfaces. In this case, the largest fraction of ELVOC would still be lost to the chamber walls (Fig. 1). In contrast, when adding seed particles, the typical condensation sink was $10 \times 10^{-3}$ s$^{-1}$. This corresponds to a lifetime of only 100 seconds with respect to the condensation to particle surfaces. This means that by introducing inorganic seed aerosol, we can change the dominating loss term of ELVOC from their condensation to the chamber walls to their condensation to aerosol surfaces, and at the same

time decrease their lifetime in the gas phase by around 60 % (Fig. 1). Assuming that the source term remains unchanged, the decreased steady state lifetimes are directly reflected in the gas phase concentrations of the ELVOC (Eq. (6)). This drop of



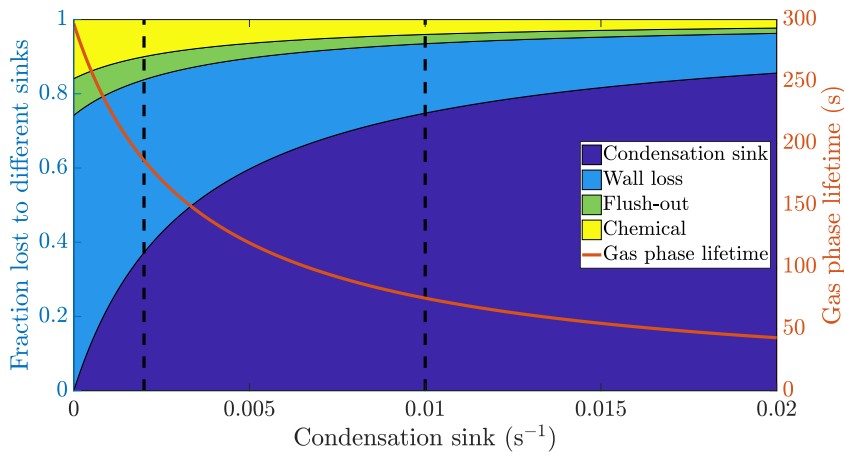

**Figure 1.** The calculated fraction of ELVOC lost to different sinks, and their total lifetime in the gas phase as a function of the condensation sink caused by aerosol particles in the chamber. The wall loss lifetime of ELVOC is assumed to be 400 seconds. The chemical loss is an upper limit estimate, based on an OH concentration of around 0.1 ppt and collision limited reaction with ELVOC. The vertical broken lines at $0.002$ s$^{-1}$ and $0.01$ s$^{-1}$ represent a typical situation without seed particles and with seed particles, respectively (Table A1).

around 60 % corresponds to a case when there is only negligible evaporation of the oxidation products back from the particle phase in the time scale of the chamber lifetime. In addition to ELVOC, this may include low volatility organic compounds (LVOC) from the lower end of their volatility spectrum. Thus, their net loss by condensation is limited by their molecular diffusion to the particle and wall surfaces, not equilibrium partitioning. If the source term of a given compound is unchanged,

this represents an upper limit for how much the seed addition can drop the gas phase signal. The exact magnitude of this drop varies from experiment to experiment, since there is some variability in the condensation sink both with and without seed aerosol (Table A1).

Based on the example cases of IVOC and ELVOC, we can outline how seed injections affect oxidation products of different volatilities. In the case of IVOC, the volatility of the product is high enough that there is negligible net condensation. Thus,

the sink of the compound is unchanged upon seed addition. Assuming that the source of the compound stays constant, the seed injection has no effect on the gas phase concentration of the compound. For ELVOC, the gas-to-particle conversion is irreversible. Upon seed injection, the condensation into aerosol particles becomes the main sink of ELVOC, and the gas phase lifetime of ELVOC decreases by around 60 % (Fig. 1). If the ELVOC source remains constant, this decrease of the lifetime results in a 60 % drop in the gas phase concentration of ELVOC. For compounds with volatilities between these extremes, such

as semi-volatile organic compounds (SVOC), the gas phase concentration can be affected, but not as much as for ELVOC. In order to assess the exact effect of the volatility of an oxidation product on its behaviour upon seed injection, we performed model simulations with the ADCHAM model, explained in more detail in Sect. 2.5.



Above, we have assumed that the source term of oxidation products stays constant upon seed injection. In the following section, we will present two important cases when this assumption does not hold, and discuss their effect on the method and the results. The first case is related to the loss of $RO_2$ radicals, important intermediates in the $\alpha$-pinene oxidation, to seed surfaces. The second case is related to the production of compounds on the chamber walls or in aerosol particles, and their subsequent evaporation to the gas phase.

### 2.4.2 Dynamics of organic peroxy radicals in the chamber

An important class of intermediates in the formation of HOM from the ozonolysis of $\alpha$-pinene are organic peroxy radicals ($RO_2$). These compounds are reactive, both towards other $RO_2$ and radicals such as NO and $HO_2$, as well as through uni-molecular decomposition (e.g. Ehn et al., 2014; Jokinen et al., 2014). If their chemical lifetime is long enough that their loss to particle surfaces becomes a non-negligible sink for them, they can be affected by the seed addition. This is an important consideration for our study. With the injections of inorganic seed aerosol, we wish to alter the sink of non-volatile oxidation products, without affecting their source. Under the conditions in our chamber, and using the reaction rate coefficients used by Ehn et al. (2014) to describe HOM formation, a lifetime of the $RO_2$ radicals of around 10 seconds can be expected. With such a chemical lifetime, the condensation of $RO_2$ on aerosol particle surfaces becomes a non-negligible, but minor, sink. During a typical seed injection, the gas phase lifetime of $RO_2$, and thus their concentration, are expected to drop by no more than 20 % (Fig. A1). This would in turn decrease the source term for all closed shell oxidation products formed in the reactions of the $RO_2$ radical. However, compared to the 60 % reduction in the gas phase lifetime of the oxidation products upon seed addition (Fig. 1), the reduction in the source rate would be a minor effect. Still, it is important to keep in mind that any changes observed in the concentrations of gas phase oxidation products during seed injections can be caused by changes to their sources, their sink, or a combination of both.

### 2.4.3 Effect of heterogeneous chemistry on the gas phase

So far we have only explicitly considered the formation of compounds in the gas phase oxidation. However, particle phase processes can potentially affect the gas-phase concentrations of compounds as well. A compound X, after condensation, can be chemically transformed in the particle phase. If the resulting product, Y, is sufficiently volatile, it can evaporate back to the gas phase:

$$X(g) \rightarrow X(p) \rightarrow Y(p) \rightarrow Y(g). \tag{7}$$

This process can affect the gas phase concentrations in two ways. First, the concentration of compound Y can increase during a seed injection due to the process. Secondly, in addition to the effect of volatility, there is a chemical sink for compound X in the particle phase. This results in less evaporation back to the gas phase than would be expected based on volatility alone, and a larger decrease in gas phase concentration. This would in turn be interpreted as a lower than actual volatility. We cannot readily distinguish between chemical reaction driven uptake and physical condensation due to low saturation vapour pressures.





However, in the experiments with crystalline ammonium sulfate particles, we assume particle phase reactions to be very slow. In contrast, the use of ammonium bisulfate and deliquesced seed particles were in part motivated by particle phase reaction enhancement (e.g. Riva et al., 2019). Thus, by comparing the different experiment types, we can gain insight into the different uptake mechanisms. Still, we cannot fully exclude the effect of particle phase processes on the response of HOM to seed

addition.

## 2.5   Modelling the behaviour of HOM in the chamber upon seed injection

In order to quantitatively relate the volatilities of the formed oxidation products to the behaviour of their gas phase concentrations under the seed injection, we performed a series of simulations using the ADCHAM model (Roldin et al., 2014). The gas phase chemistry in ADCHAM was simulated using the Master Chemical Mechanism v3.3.1 $\alpha$-pinene chemistry (Jenkin

et al., 1997; Saunders et al., 2003) and the recently developed Peroxy Radical Autoxidation Mechanism (PRAM) (Roldin et al., 2019). In short, we used the measured temperature, relative humidity, concentration of ozone and $\alpha$-pinene and chamber flow rates as input to the model. The inorganic seed aerosol was represented by a particle number size distribution similar to the one used in the experiments. Further, the modelled AS or ABS particle mass concentration was kept identical to the one measured with the AMS. This was achieved by, for every model time step, adding new seed particles to the chamber in an amount equal to

the concentration difference between the modelled dry seed particle mass, from the previous time step, and the measured mass, from the present time step. The modelled steady state particle number size distribution, without seed particles, was evaluated against the DMPS observations and optimized by tuning the new particle formation rate of particles with an initial diameter of 1.5 nm. The SOA formation in the model was represented by treating all organic vapours with a saturation concentration ($C^*$) < $10^3$ $\mu$g m$^{-3}$ as potentially condensable. The pure-liquid saturation vapour pressures ($p_0$) of the organic vapours were

calculated using the SIMPOL functional group contribution method (Pankow and Asher, 2008). In addition to the condensable vapour from the gas-phase mechanism, we also introduced 15 oxidation products of predetermined $C^*$ in the range $10^4$ to $10^{-3}$ $\mu$g m$^{-3}$, all having a fixed $Q_{\text{chemical}}$ equal to $10^{-5}$ molec cm$^{-3}$ s$^{-1}$. By tracking the behaviour of the relative concentration drop of these model compounds, before and after the seed injections, we could connect this to representative $C^*$, for conditions similar to the actual chamber experiment.

The reversible wall losses of the condensable vapours were modelled using the method proposed by Matsunaga and Ziemann (2010). In this method, the vapour loss rate to the chamber walls and the evaporation of the same vapours from the walls back to the gas-phase are represented by two different first order rate coefficients $k_{\text{wall}}$ and $k_{\text{wall,back}}$. For each individual condensable organic compound ($i$), $k_{\text{wall}}$ was estimated using Eq. (8) and $k_{\text{wall,back}}$ with Eq. (9):

$$k_{\text{wall}, i} = \frac{1}{400} \sqrt{\frac{D_i}{D_{\text{C}_{10}\text{H}_{16}\text{O}_7}}}, \tag{8}$$

$$k_{\text{wall, back}, i} = k_{\text{wall}, i} \frac{p_{0,i}}{C_{\text{wall}} RT}. \tag{9}$$





The Teflon walls are treated as a large organic aerosol concentration ($C_{\text{wall}}$), which absorb the organic vapour molecules that hit the walls. We used a $C_{\text{wall}}$ equal to 100 $\mu$mol m$^{-3}$, which is within the range of values reported by Matsunaga and Ziemann (2010). $D_i$ and $D_{\text{C}_{10}\text{H}_{16}\text{O}_7}$ in Eq. (8) are the molecular diffusion coefficients for compound $i$ and the reference HOM molecule $\text{C}_{10}\text{H}_{16}\text{O}_7$, respectively. $R$ is the ideal gas constant (8.3145 J K$^{-1}$ mol$^{-1}$) and $T$ is the temperature in Kelvin. The organic

compound molecular diffusion coefficients were calculated with Fuller's method (Tang et al., 2015). Equation (8) takes into account that large organic molecules have a slower diffusion than small molecules and therefore lower $k_{\text{wall}}$. As an example, for a HOM dimer with molecular formula $\text{C}_{20}\text{H}_{30}\text{O}_{16}$ the first order wall loss rate becomes 1/481 s$^{-1}$ (17 % lower than for $\text{C}_{10}\text{H}_{16}\text{O}_7$).

### 2.6   Interpretation of the CI-APi-TOF data

The vast majority of the ions detected with the CI-APi-TOF were clusters of analyte molecules with the nitrate ion, $\text{NO}_3^-$. However, a minor fraction appeared to be analyte molecules clustered with the dimer of nitric acid, $(\text{HNO}_3)\text{NO}_3^-$, as seen before by e.g. Bianchi et al. (2016, supplementary information). Analyte molecules that do not contain nitrogen, clustered with the dimer of nitric acid, appear at exactly the same masses as some organic nitrates clustered with only the nitrate ion. However, they are seen even without nitrogen oxides ($\text{NO}_{\text{X}}$) in the chamber, when no organic nitrates are formed. Further proof of their

identity as non-nitrates clustered with the nitric acid dimer comes from the good correlation of their time behaviour with that of the corresponding peaks clustered only with the nitrate ion. Due to this, for the analyses during the experiments without $\text{NO}_{\text{X}}$ in the chamber, we excluded any peaks containing more than one nitrogen atom, i.e. the one from the nitrate ion. In the presence of $\text{NO}_{\text{X}}$, we observed a large increase in the signals at these masses, attributed to the formation of organic nitrates appearing at the same masses. In these experiments, the organic nitrates probably contributed the clear majority to these signals, but there

was a non-zero contribution from the dimer charging as well. This should be taken into account when interpreting the results. In general, we observed that the dimer clustering seemed to be more prominent with the LTOF instrument used here, compared to the older high resolution APi-TOFs (HTOFs).

In addition to the analyte molecules charged with the nitrate ion or its dimer, some molecules are also detected as deprotonated anions. For simplicity, we excluded these peaks from the analyses.

$\text{RO}_2$ radicals formed in the ozonolysis or OH oxidation of $\alpha$-pinene, and organic nitrates containing one nitrate group, both have an odd number of hydrogen atoms, causing them to appear at odd masses in the spectra. Further, the difference in mass is often very small. This causes problems for the separation of the signals originating from $\text{RO}_2$ radicals from those coming from organic nitrates (or non-nitrate compounds charged with the dimer of nitric acid, as noted above). The signals may not be unambiguously separated, and the signal attributed to one may have some contribution from the other. For this reason, we

do not present detailed results on the behaviour of $\text{RO}_2$ radicals during seed injection. Organic nitrates often have much higher signals as compared to $\text{RO}_2$ radicals, which makes their fits more robust (Cubison and Jimenez, 2015). Due to this, we do present results on their behaviour. However, due to the overlap with some $\text{RO}_2$ radicals, along with the dimer charging effect discussed above, these results should be interpreted with some caution.



### 2.7 Data processing

We processed the nitrate CI-APi-TOF data using tofTools (Junninen et al., 2010). After high resolution peak fitting, we normalized the signal intensities by the sum of the reagent ion signals, taking into account the nitrate monomer, dimer and trimer signals. We then used these normalized signals in subsequent analysis. As the analysis focuses on relative changes in signal intensities, absolute concentration calibrations were not necessary.

In order to investigate the effect of seed injections on gas phase concentrations of $\alpha$-pinene oxidation products, we compared the signal levels during the seed injections to those during the steady state before the injection. For assessing the reliability of the signals, we calculated the mean signal level during the steady state before the seed injection, along with the standard deviation (SD) of the signal during this period. A high standard deviation in relation to the mean signal level can mean either a noisy signal, or an unstable one. For this reason, we excluded the compounds with a SD-to-mean ratio above four from further analyses.

## 3 Results

### 3.1 General behaviour of HOM and organic aerosol upon seed injections

When injecting only gaseous precursors we observed formation of both HOM monomers and dimers, as expected. The HOM formation was accompanied by the formation of SOA. During the course of an experiment, both the HOM signals measured by the CI-APi-TOF, and the organic mass measured by the AMS, stayed stable, indicating they were in steady state ($SS_1$ in Fig. 2). After sampling this steady state for several hours, we started injecting inorganic seed aerosol. The seed aerosol concentration reached a steady level after a few hours ($SS_2$ in Fig. 2). This increase in the seed aerosol concentration resulted in an increased condensation sink, leading to enhanced condensation of HOM, and a corresponding increase in the organic aerosol mass (Fig. 2). The behaviour of both gas-phase HOM, as well as the SOA, were well captured with the ADCHAM model (Fig. A2).

### 3.2 Expected relationship between vapour pressure and condensation behaviour of HOM

Using the ADCHAM model, we found that in the conditions of the chamber, the gas phase concentrations of oxidation products with a saturation concentration ($C^*$) over 100 $\mu$g m$^{-3}$ (in the volatile end of the SVOC range) are not expected to be affected by the seed additions due to fast evaporation. Thus, the experimental setup cannot readily give precise information on the volatilities of compounds having saturation concentrations above this value. At the other extreme, products with saturation concentrations below 0.01 $\mu$g m$^{-3}$ (within the LVOC range) are expected to all show a behaviour consistent with irreversible condensation. Between these limiting cases, there is a smooth transition across the SVOC and upper end of LVOC range: it is in this area that the method can give the most precise information on volatility. (Fig. 3).

The response of the HOM to the seed injection was not uniform: some compounds showed a larger fractional decrease than others. As an example, only a small fraction of the original concentration of $C_{10}H_{14}O_9$ remained in the gas phase, while the concentration of $C_{10}H_{14}O_6$ was almost unaffected (Fig. 2). To better assess the exact magnitude of the decrease, we normalized



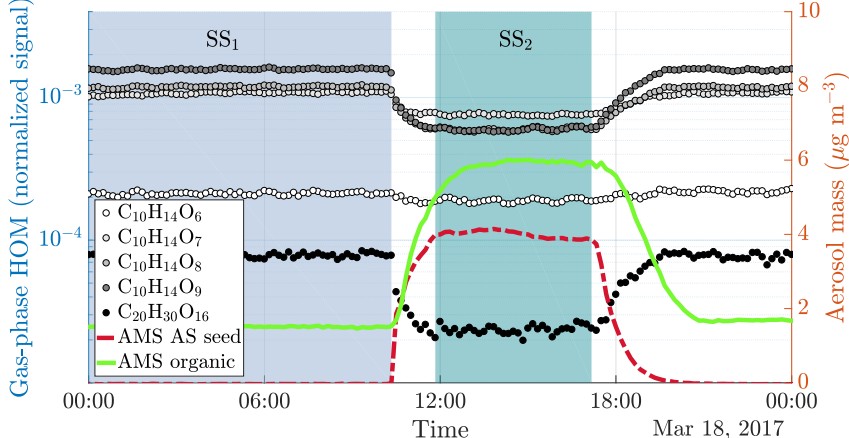

**Figure 2.** Time series of both gas and particle phase species during a typical seed injection (Experiment 19, Table A1). The experiment starts with only gaseous precursors being injected into the chamber: these form HOM (dotted lines, measured with the $NO_3^-$-CI-APi-TOF), which form SOA (solid green line, measured with the AMS). Both the HOM and the SOA are in a steady state ($SS_1$, green shading). At around 10 o'clock, we start injecting ammonium sulfate seed aerosol (red dashed line). This results in an increased condensation sink, which causes an increase in the SOA mass, and a decrease in the gas phase HOM signals. After a transition period, a new steady state is reached ($SS_2$, red shading). Upon stopping the seed injection, the SOA levels decrease again, and the HOM signals increase.

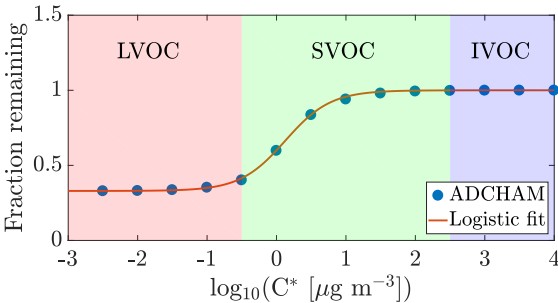

**Figure 3.** Modelled fraction remaining vs. the logarithm of saturation concentration from the ADCHAM model for experiment number seven (Table A1). In addition to the discrete data points representing model compounds in ADCHAM (Sect. 2.5), a logistic fit to them, with the formula $y = \frac{1-0.33}{1+e^{-3.0(x-0.13)}}$, is shown. Shading according to the volatility classes by Donahue et al. (2012).

the gas phase signal of each identified compound in the CI-APi-TOF spectrum to its level during the steady state before the seed injection (Fig. 4). In the experiment shown, the gas phase concentration of $C_{10}H_{14}O_6$ decreased only by around 10 % (Fig. 4). This decrease can be explained by a saturation concentration of around 5 $\mu g\ m^{-3}$, which lies well within the SVOC range. In contrast, the concentration of the more highly oxygenated $C_{10}H_{14}O_9$ decreased by more than 60 %, indicating a

5    saturation concentration of around 0.2 $\mu g\ m^{-3}$, on the volatile end of LVOC range. The analogous compounds with seven and





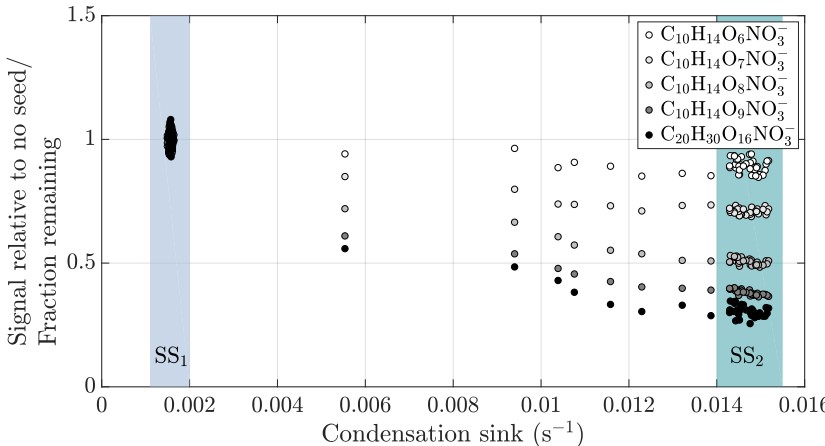

**Figure 4.** Gas phase HOM signal normalized to level before seed injection vs. condensation sink during a typical experiment (Experiment 19, same as in Fig. 2). The steady state before seed injection ($SS_1$) is visible as a cluster of points having a low condensation sink, and normalized gas phase signals of around 1. The steady state formed during the seed injection ($SS_2$) is seen as distinct clusters of points at higher condensation sink: the transition between the steady states is visible between these clusters. For each compound separately, we averaged the values during the seed injection steady state, and used these average values in the subsequent analyses.

eight oxygen atoms exhibited behaviours between the two, showing a progression towards larger decreases with increasing oxygen number. This progression can be expected if we expect that the volatility of HOM gradually decreases with increasing level of oxygenation. In comparison to the monomers, the decrease in the gas phase concentration of $C_{20}H_{30}O_{16}$, an example of an essentially non-volatile HOM dimer, was slightly greater in magnitude to the decrease of $C_{10}H_{14}O_9$. This decrease of

around 70 % is consistent with kinetically limited condensation, and behaviour expected of essentially non-volatile vapours. It also demonstrates the loss of sensitivity to large changes in volatility below around 0.3 $\mu$g m$^{-3}$: the saturation concentration of $C_{20}H_{30}O_{16}$ is presumably orders of magnitude smaller than that of $C_{10}H_{14}O_9$, yet they differ only slightly in their condensation behaviour.

### 3.3   Condensation behaviour of HOM as a function of molecular mass and composition

We will next analyze the behaviour of the measured gas phase compounds in experiments conducted in a dry chamber, without $NO_X$, using ammonium sulfate seed particles. We will then compare these results to those measured in the presence of $NO_X$, in a humid chamber, and using ammonium bisulfate instead of ammonium sulfate. To facilitate the analysis, we averaged the values for the fraction remaining for each fitted compound during the seed injection steady state, as demonstrated in Fig. 4, for each experiment separately. We then used these steady state average values for the fraction remaining in subsequent analyses.

Looking at the fraction remaining after seed injection as a function of the mass of the detected cluster, we generally observed values close to one for many of the lower mass compounds, up to 250 Da (including 62 Da from the charging nitrate ion) (Fig. 5). This indicates that the volatilities of these compounds generally fall into the volatile end of the SVOC, or IVOC range, or

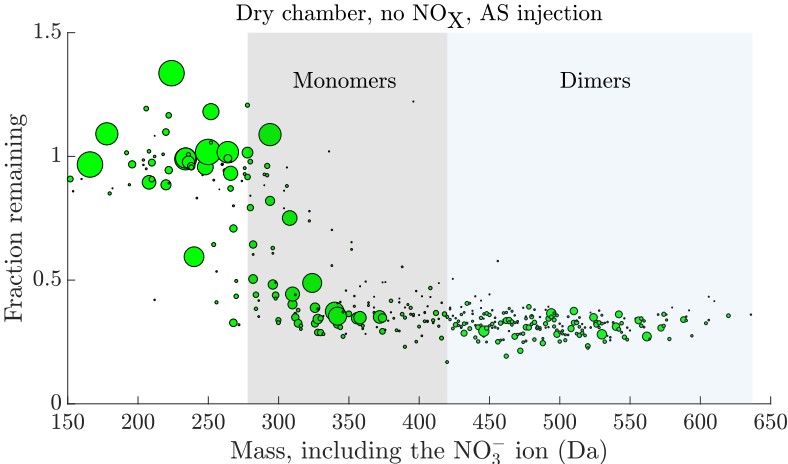

**Figure 5.** Molecular mass of the detected cluster vs. the fraction remaining after seed injection. The area of the circles has been scaled to the magnitude of the signal of each compound before the seed injection, capped at $3.2 \times 10^{-3}$. The data are the average of experiments 2 and 12, with ammonium sulfate injection, dry chamber, and no $NO_X$ in the chamber. Compounds with a signal to noise ratio, as defined in Sect. 2.7, above 4 have been excluded from the plot. Characteristic HOM monomer and dimer regions have been highlighted with grey and blue shading, respectively.

higher (Fig. 3). At the other end of the spectrum, we observed values around 0.3, indicating volatilities in the LVOC range or lower, for most high mass compounds starting from 350 Da (Fig. 5). Between the masses 250 Da and 350 Da there was a gradual, but consistent shift from the values close to 1 to values close to 0.3. This transition had a distinct sigmoidal shape, much like the transition of the expected fraction remaining with decreasing volatility, based on the ADCHAM model (Fig. 3).

At the masses below 250 Da, we detect molecular fragments with a carbon skeleton shorter than that of the precursor $\alpha$-pinene, and relatively few oxygen atoms. Due to their short carbon skeletons and low level of oxygenation, they are expected to be relatively volatile. On the other hand, on masses above 350 Da, we mainly detect the most oxidized HOM monomers with more than 10 oxygen atoms, along with HOM dimers. These compounds are expected to have low, or even extremely low, volatility (Ehn et al., 2014; Kurtén et al., 2016; Tröstl et al., 2016; Bianchi et al., 2019). Between these extremes the sigmoid-

like transition of compounds is consistent with a gradual decrease of volatility with increasing mass, coinciding with increased levels of oxygenation of the compounds. This transition seems to be centered around 310 Da: around this mass, the highest signals seem to be around half way between the high- and low volatility limits, i.e. with a saturation vapour concentration of around 1 $\mu$g m$^{-3}$ (Fig. 3). The behaviour of the compounds during the seed injection thus seems consistent with what we would expect based on the presumed volatilities of the oxidation products. However, as stated in Sect. 2.4, we cannot exclude

the possible role of a changed source term, or reactive uptake, for some of the compounds in determining the change in their gas phase concentrations during the seed injection. Still, we expect that the volatility of the compounds is dominant in determining their behaviour.





In addition to the closed shell products, we also investigated the effect of the seed additions on the highly oxidized $RO_2$ radicals measured with the CI-APi-TOF. However, due to the limitations listed in Sect. 2.6, we will not go into detail regarding the the behaviour of $RO_2$. Nonetheless, we observed some decrease in the signals attributed to them during the seed injections, but these decreases generally did not exceed 20 % of their signal before the seed addition. This further supports the idea that

the majority of the changes observed in the closed shell species upon seed addition are indeed caused by differences in their volatility, and not in the differences in their source terms. It seemed that the decrease in the concentrations of individual $RO_2$ was not entirely uniform, potentially indicating differences in their lifetimes and/or formation dynamics. We are investigating this phenomenon, and its effect on the formation of closed shell HOM, in more detail, and will publish these results separately.

### 3.4 Effect of $NO_X$

Upon introduction of $NO_X$ in the chamber, we observed the formation of many oxidation products containing nitrogen (Fig. 6 (a)). These compounds are expected to form when $RO_2$ radicals are terminated by NO (Ehn et al., 2014), or when the oxidation is initiated by the nitrate radical (Yan et al., 2016): both result in the oxidation products containing an organic nitrate functional group. Compared to the experiments without $NO_X$, the behaviour of the non-nitrate oxidation products did not seem to change significantly (Figs. 5 and 6 (a)). In contrast, compared to the oxidation products consisting of only carbon, hydrogen, and

oxygen, the organic nitrates seemed to transition to low volatilities at markedly higher masses, with the transition centered around 355 Da (Fig. 6 (a), again including 62 Da from the charging nitrate). This is around 45 Da higher than for the non-nitrates. This difference in mass for compounds of similar volatility is to be expected based on the functional groups making up these oxidation products. Non-nitrate HOMs are expected to contain carbonyl, hydroperoxide, hydroxide, peroxyacid and carboxylic acid groups, in agreement with known reaction pathways for peroxy radicals (Ehn et al., 2014). The organic nitrate

HOMs contain the nitrate group as well, acquired in the radical termination reaction with NO (or the initiation of the oxidation with $NO_3$). In the absence of NO, many termination pathways of $RO_2$ radicals lead to the formation of a hydroxy group (Vereecken and Francisco, 2012). While the nitrate group has a mass of 62, the hyrdoxyl has a mass of only 17 Da. Despite their mass difference, they are expected to lower the volatility of a compound by roughly the same amount (Kroll and Seinfeld, 2008; Pankow and Asher, 2008). Hence, for two otherwise identical compounds, one containing a hydroxy group and the other

a nitrate group, a similar volatility can be expected, while the organic nitrate has a mass 45 Da higher. This is analogous with a situation where a peroxy radical gets terminated either through reaction with NO or with $HO_2$, as an example: the resulting oxidation products show a large difference in mass, but have similar volatilities. Of course, these are not the only possibilities for the formation of organic nitrates and non-nitrates, but highlight that the mass difference for compounds of similar volatility is to be expected.

### 3.5 Effect of seed composition and chamber humidity on the condensation behaviour of HOM

Compared to the injections of ammonium sulfate, we did not find a large difference in the behaviour of the gas phase oxidation products measured with the $NO_3^-$-CI-APi-TOF upon injections of the more acidic ammonium bisulfate. In contrast, we did observe a difference between the experiments conducted at < 1 % RH with effloresced seed, and those at 40 % RH with



(a)

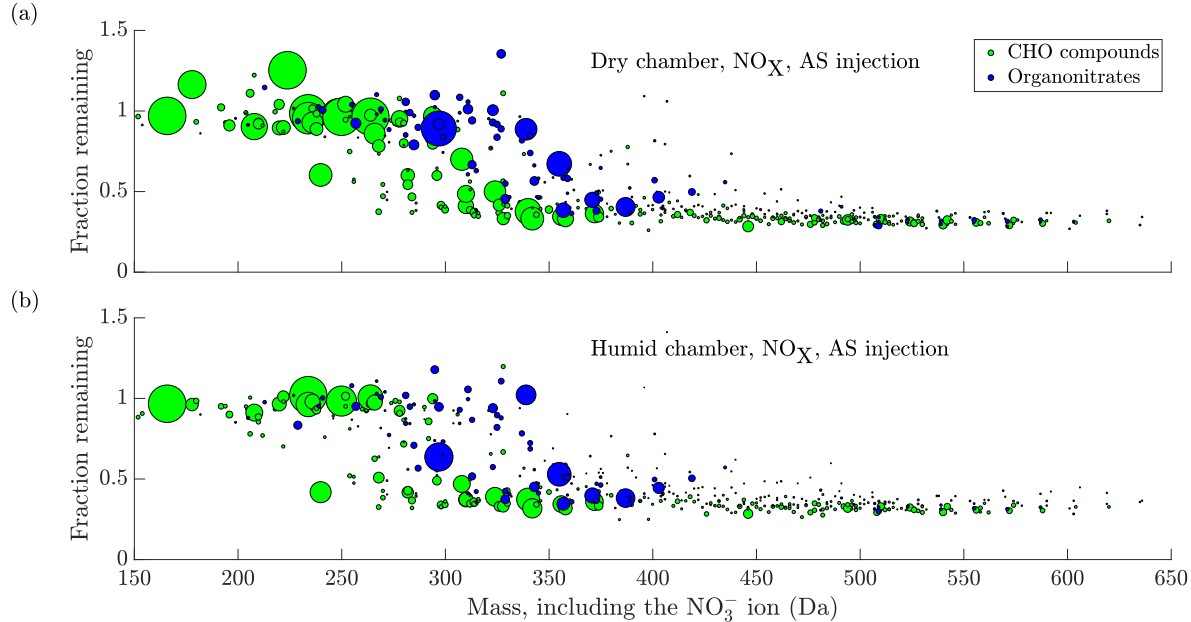

**Figure 6. (a)** Molecular mass of the detected cluster, including the charging $NO_3^-$ ion, vs. the fraction remaining after seed injection. Area of the circles scaled to the magnitude of the signal of each compound before the seed injection, capped at $3.2 \times 10^{-3}$. The data are the average of the experiments 16 and 19, with ammonium sulfate injection to dry chamber, in the presence of $NO_X$. Compounds with a signal-to-noise ratio, as defined in Sect. 2.7, above 4 have been excluded from the plot. **(b)** same as (a), but in humid conditions (experiment 18, 40 % RH and deliquesced seed).

deliquesced seed, with many compounds seeming to decrease more during seed addition in the humid chamber (Fig. 6 (b)). It is plausible that this increased uptake of the oxidation products was caused by the formation of an aqueous phase on the particles, and solubility-driven or reactive uptake of the oxidation products to the aqueous phase.

## 3.6 Factors determining HOM volatility

5  To gain more insight on what determines the volatility of HOM, we constructed a statistical model explaining their condensation behaviour, measured by the fraction remaining, in terms of their composition. From Fig. 6, it is already clear that the molecular mass of the compounds explains the volatility relatively well, with increasing mass decreasing the volatility. However, molecular mass cannot explain all the features of the curve, such as the different behaviour of nitrates and non-nitrates. Furthermore, the molecular mass in itself provides no causal explanation for the volatility: rather, low volatility is caused by

10  intermolecular forces: thus, intermolecularly bonding functional groups lower the volatility of a compound (e.g. Kroll and Seinfeld, 2008). These are not directly measured by the mass spectrometric technique, but the composition of a compound gives some indication as to what functional groups it may contain. As an example, a hydroxy group will contribute, in addition





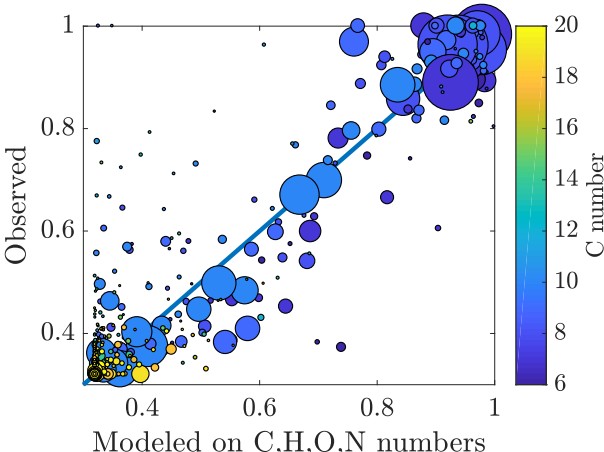

**Figure 7.** Measured fraction remaining of the y-axis vs. fraction remaining modelled on the chemical composition. The colouring of the circles is based on the carbon number of the compound, and the size of the circles is scaled according to the signal intensity, as in Figs. 5 and 6. The coefficients for the binomial model are presented in Eq. (10). Number of compounds used for the fit: n = 442, Chi$^2$-statistic vs. constant model: $1.81 \times 10^4$, p-value = 0 (to machine precision). A good correspondence is found between the measured and fitted values, indicating that the volatility of a compound can be explained well in terms of its molecular formula.

to a hydrogen atom, one oxygen, while a hydroperoxide group will contribute two. In contrast, an aldehyde or a ketone group will contribute one oxygen, while reducing the number of hydrogen atoms bound to the carbon skeleton by one. Thus, all of these groups will contribute a different number of hydrogen and oxygen atoms. However, these contributions are not unique, as a compound with two hydroxy groups will have the same composition as that having one hyrdoperoxide group, everything else being equal. Still, the composition can provide indication of the functionalities of a molecule as well.

### 3.6.1 Logistic regression to explain HOM volatility

For the analysis, we linearly scaled the fractions remaining (FR) of HOM to range from zero to one, with the upper branch of the sigmoid in Fig. 6 getting the value one, and the lower branch, containing essentially non-volatile HOM dimers, the value zero. We used this scaled response (FR$_\text{scaled}$) as the dependent variable, with the carbon, hydrogen, oxygen and nitrogen numbers of a molecule as the independent variables. With the dependent variable ranging from zero to one, and having a sigmoidal transition between the extremes, we chose to use a generalized linear model with a logit link function. For the modeling we chose to use the average of experiments 16 and 19 (Table A1), with NO in the chamber, as in this way we could incorporate organic nitrates into our model. Compounds with less than six carbon atoms, or with a signal to noise ratio below four, were excluded from the model, in order to avoid the smallest fragments and unreliable signals, respectively. In addition, the model was weighted with the signal before seed addition, capped to the same value as in Fig. 6, in order to give more weight to the more abundant oxidation products.

We found that the fraction remaining could be well explained using the composition (Fig. 7). The formula for the fit is:





$$\text{logit}(\text{FR}_{\text{scaled}}) = 0.5 \times n_{\text{C}} - 0.4 \times n_{\text{H}} - 1.1 \times n_{\text{O}} + 2.3 \times n_{\text{N}} + 8.7. \tag{10}$$

The coefficients for each of the terms in the model, including the intercept, were highly statistically significant, with all p values being smaller than $10^{-53}$. To convert the dependent variable to volatility, we used the logistic fit between the saturation concentration of a model compound, and the its expected fraction remaining from Fig. 3. By combining that relationship with the logistic fit in Eq. (10), we can obtain an expression for the saturation vapour concentration in terms of the composition:

$$\log_{10}(\text{C}^*[\mu\text{g m}^{-3}]) = 0.16 \times n_{\text{C}} - 0.13 \times n_{\text{H}} - 0.36 \times n_{\text{O}} + 0.76 \times n_{\text{N}} + 2.8. \tag{11}$$

The addition of a carbon or a nitrogen atom to a molecule, with their positive coefficients in the model, would act as to increase the volatility. The addition of a hydrogen or an oxygen atom, on the other hand, would act as to decrease it. The positive coefficients of carbon and nitrogen seem counter-intuitive at first. However, the addition or removal of a carbon or a nitrogen atom is not independent of other elements. In the case of HOM formed here, the nitrogen most probably appears in the form of a nitrate functional group ($\text{NO}_3$), and is thus accompanied by three oxygen atoms. Adding the coefficients of the nitrogen atom with three oxygen results in a negative overall coefficient. The cumulative effect of adding a nitrate group would therefore act to reduce the fraction remaining, and thus volatility, in agreement with widely used group contribution methods, such as SIMPOL developed by Pankow and Asher (2008). Similarly, addition or removal of a carbon atom, while keeping the structure of the molecule otherwise unchanged, would be accompanied by the corresponding gain or loss of two hydrogen atoms. The combined effect would still be to decrease the volatility with increasing carbon number, as expected. Even accounting for the carbon number in the model, the hydrogen number of a molecule also got a statistically significant coefficient. This indicates that the hydrogen number provides independent information on the volatility of a compound, with the volatility decreasing with increasing hydrogen number. The termination step of an $\text{RO}_2$ radical formed in the ozonolysis of $\alpha$-pinene determines the number of hydrogen atoms in the resulting molecule, with lower hydrogen numbers being accompanied by the formation of a carbonyl group, and higher numbers accompanied with a hydroxy group (Vereecken and Francisco, 2012). The hydroxy group is expected to lower the volatility of a compound more than the carbonyl group (Kroll and Seinfeld, 2008). Thus, the negative coefficient of hydrogen in our model, as well as the coefficients of other elements, are reasonable in the light of molecular considerations for the determination of volatility.

As before, any compounds with a saturation concentration higher than 100 $\mu\text{g m}^{-3}$ (volatile end of SVOC range) are expected to behave in the same way, not being affected by the seed addition, and showing values for fraction remaining of around one (Fig. 3). Similarly, compounds with a saturation concentration below 0.01 $\mu\text{g m}^{-3}$ (in the LVOC range) are all expected to behave as non-volatile. Thus, the model, similarly to the measured results, cannot distinguish between volatilities of compounds above and below these limits, respectively. From this follows that the model is most reliably fit to the compounds in the transition region, in the SVOC to LVOC range. The relationship given in Eq. (11) may only hold for that range of volatilities. Thus, caution should be used in interpreting the modelled volatilities, especially outside the range of volatilities





defined above. E.g. HOM dimers, which all exhibit behaviour consistent with non-volatile vapours, are correctly fit in the very lowest edge of the volatility range, but any detailed information on their volatility is lost. For this reason, the relationship given in Eq. (11) may be better suited for HOM monomers. Also, the relationship presented in Eq. (11) may not hold for oxidation systems other than the one presented here. Therefore, any generalizations should be drawn with caution.

### 3.6.2 Comparison to existing volatility parametrizations

There are numerous existing parametrizations for assessing the volatility of VOC oxidation products. Some, like the SIMPOL model by Pankow and Asher (2008), require as inputs the exact functional groups making up a molecule. Others, such as the one presented in Bianchi et al. (2019), and the one devised by Tröstl et al. (2016), only require the molecular formula, having some underlying structural assumptions. The former requires the carbon and oxygen numbers in the molecule. The presence of nitrogen is separately handled, by assuming it to exist in nitrate functional groups, assigning a constant effect to that nitrate group. The latter is based on the estimated SIMPOL volatilities for certain expected model HOM compounds, and fit based on their O:C ratios, separately for $C_{10}$ monomers and $C_{20}$ dimers, with each group getting their own relationship. As the exact structures of HOM are not known (Bianchi et al., 2019), any method requiring structural information, such as SIMPOL, will include some uncertainty stemming from the choice of structure for a compound. Similarly, the SIMPOL-based parametrization by Tröstl et al. (2016), will include this uncertainty.

In the SIMPOL model, any addition of oxygen to the molecule decreases its volatility by almost an order of magnitude, at minimum (for a ketone). A hydroxy or a hydroperoxy group both lower the volatility by over 2 orders of magnitude, as does a nitrate group. In our parametrization (Eq. (11)), the coefficient for oxygen is $-0.36$. This means that with the addition of one oxygen atom, the volatility of a compound is reduced by only a bit more than a factor of 2. Thus, even for a hydroperoxy group, containing two oxygens, the reduction in volatility would be only 5-fold. For a nitrate group, the corresponding decrease in volatility would be a factor of 2. Thus, our parametrization in Eq. (11) seems to predict much smaller sensitivities of the volatility of HOM to different functional groups than what would be expected based on e.g. SIMPOL.

Using quantum chemical calculations by the COSMO-RS model, Kurtén et al. (2016) found that intramolecular hydrogen bonding between the functional groups within a HOM molecule may inhibit the ability of the groups to take part in intermolecular bonding, and thus to reduce the volatility of the molecule. This would lead to the volatility being less sensitive to addition of functional groups than would be expected based on e.g. SIMPOL. As noted above, our results support this lower than expected sensitivity.

As a result of the lower sensitivity of the volatility to additional functional groups, Kurtén et al. (2016) hypothesised that the volatilities of HOM may be higher than expected based on group contribution methods. As a best estimate, they suggested to use the geometric mean of the SIMPOL and COSMO-RS values for the volatility of HOM. Additionally, Kurtén et al. (2016) found that for the same molecular formula, the volatility estimates by both the group contribution methods and by the COSMO-RS model varied up to 4 orders of magnitude depending one the exact structure chosen. To compare our results to those presented by Kurtén et al. (2016), as well as the parametrizations by Bianchi et al. (2019) and Tröstl et al. (2016), we





**Table 1.** Comparison of different volatility estimates and parametrizations. SIMPOL and COSMO-RS are from Kurtén et al. (2016): COSMO-RS is the geometric mean of their four different COSMO-RS estimates. Geom. mean is the geometric mean of SIMPOL and COSMO-RS as recommended by Kurtén et al. (2016). For compounds with six to eight oxygen atoms, Kurtén et al. (2016) used multiple candidate isomers: values are given separately for the one with the highest (subscript h) and lowest (subscript l) saturation concentration. For the parametrizations from Tröstl et al. (2016), Bianchi et al. (2019) and Eq. (11), all structural isomers get the same value, and thus only one is given. All values are saturation vapour concentrations in $\mu$g m$^{-3}$. The values from Kurtén et al. (2016) are calculated at 298.15 K, Tröstl et al. (2016) at 293 K, Bianchi et al. (2019) at 300 K and Eq. (11) at chamber temperature, approx. 302 K.

| molecule | SIMPOL | COSMO-RS | Geom. mean | Tröstl et al. (2016) | Bianchi et al. (2019) | Eq. (11) |
|---|---|---|---|---|---|---|
| $C_{10}H_{16}O_6$ | h$1.3\times10^1$ | h$5.1\times10^3$ | h$2.6\times10^2$ | $8.3\times10^0$ | $1.5\times10^{-1}$ | $1.4\times10^0$ |
| | l$6.0\times10^{-1}$ | l$1.0\times10^2$ | l$7.8\times10^0$ | | | |
| $C_{10}H_{16}O_7$ | h$7.2\times10^0$ | h$4.6\times10^3$ | h$1.8\times10^2$ | $4.4\times10^{-1}$ | $2.0\times10^{-2}$ | $6.3\times10^{-1}$ |
| | l$3.9\times10^{-3}$ | l$2.2\times10^1$ | l$2.9\times10^{-1}$ | | | |
| $C_{10}H_{16}O_8$ | h$2.4\times10^1$ | h$1.4\times10^3$ | h$3.1\times10^1$ | $2.2\times10^{-2}$ | $3.2\times10^{-3}$ | $2.8\times10^{-1}$ |
| | l$2.2\times10^{-3}$ | l$3.3\times10^1$ | l$2.7\times10^{-1}$ | | | |
| $C_{10}H_{16}O_9$ | $2.6\times10^{-2}$ | $1.1\times10^3$ | $5.4\times10^0$ | $1.2\times10^{-3}$ | $6.3\times10^{-4}$ | $1.2\times10^{-1}$ |
| $C_{10}H_{16}O_{10}$ | $8.7\times10^{-2}$ | $2.1\times10^1$ | $1.3\times10^0$ | $6.3\times10^{-5}$ | $1.3\times10^{-4}$ | $5.2\times10^{-2}$ |

took a set of model HOM compounds from Kurtén et al. (2016), and calculated the volatility estimates for them using both those parametrizations, as well as using Eq. (11).

For $C_{10}H_{16}O_X$, with $X$ ranging from 6 to 10, we found that our parametrization generally gives lower volatilities than the geometric mean of COSMO-RS and SIMPOL estimates, especially for the higher oxygen numbers (Table 1). However,
as noted above, both of those methods show a large variability depending on the exact structure of the molecule. However, compared to the parametrizations by Bianchi et al. (2019) and Tröstl et al. (2016), and the lower end of SIMPOL estimates, our parametrization generally gives higher volatilities. Further, as noted above, our parametrization is much less sensitive to the addition of oxygen as compared to either Bianchi et al. (2019) or Tröstl et al. (2016). In this aspect, our parametrization is closer to COSMO-RS. However, the actual volatility estimates in our parametrization are much lower than those given by
COSMO-RS. Our results thus fit in with the existing literature, in that the volatility of HOM seems to be less sensitive to oxygen addition than expected from SIMPOL, as suggested by Kurtén et al. (2016). However, the absolute values of the volatility seem to be lower than those suggested by Kurtén et al. (2016) but still higher than from Bianchi et al. (2019). Also, as noted above, we cannot fully exclude the role of particle phase processes in artificially lowering our HOM volatility estimates.

## 4 Conclusions

To investigate the volatility of HOM formed in the ozonolysis of the monoterpene $\alpha$-pinene, we used injections of inorganic seed aerosol to promote their condensation in a continuous flow chamber experiment. We found that, as expected, the general trend was that the higher the mass of the oxidation product, the more their gas phase signal dropped during the seed injections,



down to levels consistent with irreversible condensation. The observed changes were consistent with the lowering of the volatility of the compounds with increasing mass. The most highly oxidized HOM monomers, along with HOM dimers, were determined to be of low or extremely low volatility. Compared to non-nitrate oxidation products, we found that organic nitrates of comparable volatility had a higher mass, probably due to the relatively high mass of the nitrate group. The type of seed (ammonium sulfate, or the more acidic ammonium bisulfate) did not have a notable effect on the condensation behaviour of HOM, while in a humid chamber the uptake of some compounds was observed to be higher.

We found that the behaviour of the compounds upon seed injection, and thus their volatility, could be well explained in terms of their chemical composition. We found carbon, hydrogen, oxygen and nitrogen numbers all to be important in explaining the volatility, and the relationship could be connected to molecular properties of the compounds. Based on this relationship, we were able to develop a parametrization for the volatility of HOM monomers generated in $\alpha$-pinene ozonolysis. In future studies, this parametrization should be used to further clarify the role of HOM in new particle formation.

The results presented here are possibly specific to HOM from the ozonolysis of $\alpha$-pinene, but the general methodology should be applicable to other conditions as well. These conditions may include other oxidant-VOC-combinations, but also different loadings of organic aerosol to probe different volatility ranges. However, in investigations of volatility, care should be taken to affirm that the observed changes in gas phase signals are in fact caused by volatility, and not changes in gas phase chemistry, for example.

*Code and data availability.* Data will be available from a persistent repository and upon request from corresponding authors. Codes for the analysis will be available from OP, and the ADCHAM model code is available from PR upon request.

**Appendix A**

**A1**

*Author contributions.* Conceptualization: OP, MR, ME; Formal analysis: OP (lead), PR (supporting); Funding acquisition: OP, ME; Investigation: OP, MR, LH, LQ, ME; Methodology: OP, MR, PR, ME; Software: PR; Supervision: ME; Validation: OP; Visualization: OP; Writing - original draft: OP; Writing - review & editing: All coauthors.

*Competing interests.* The authors declare that they have no competing interests.

*Acknowledgements.* We would like to thank Olga Garmash and Chao Yan for helpful discussions, and Simon Schallhart for help in interpreting the PTR-TOF data. This work was funded by the European Research Council (ERC-StG 638703-COALA), the Academy of Finland



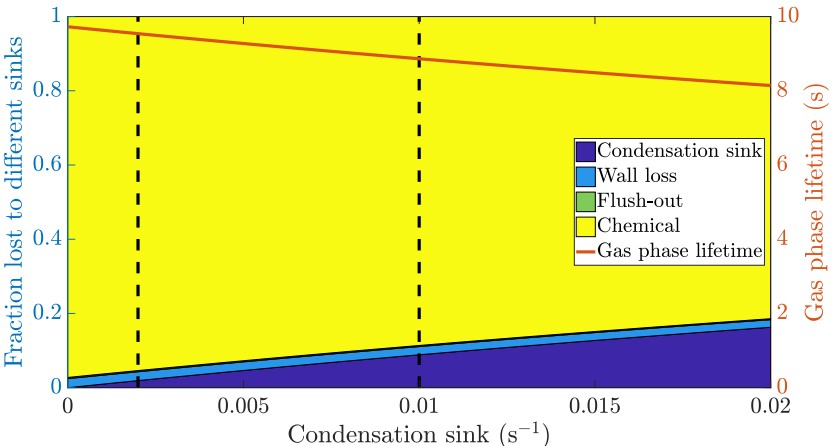

**Figure A1.** The calculated fraction of $RO_2$ radicals lost to different sinks, and their total lifetime in the gas phase as a function of the condensation sink caused by aerosol particles in the chamber. The chemical lifetime of $RO_2$ is estimated to be 10 seconds, and the wall loss lifetime 400 seconds. The vertical broken lines at 0.002 s$^{-1}$ and 0.01 s$^{-1}$ represent a typical situation without seed particles and with seed particles, respectively.

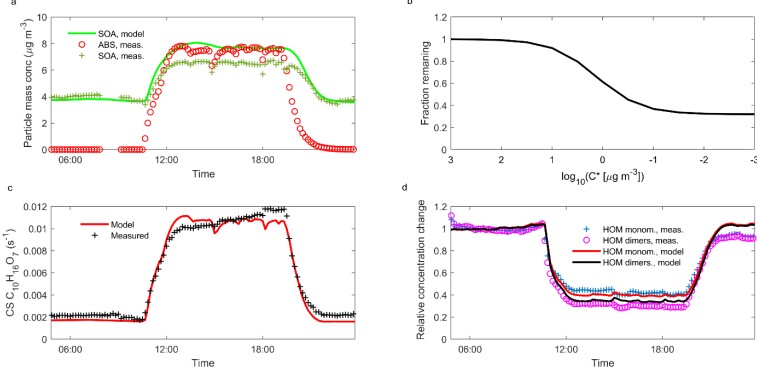

**Figure A2.** Experiment 4 modeled with ADCHAM. a: the measured and modelled ABS and organic aerosol concentrations. b: the modelled fraction remaining as a function of saturation vapour concentration. c: modelled and measured CS for a compound with molecular formula $C_{10}H_{16}O_7$. d: modelled and measured relative change in the concentration of HOM monomers and dimers. The measured HOM monomers and dimer are represented by the total concentration of molecules in the mass range 290-450 Da and 452-600 Da (including the $NO_3^-$ ion).

(Project numbers 317380 and 320094), the Vilho, Yrjö and Kalle Väisälä Foundation and the Swedish Research Council Formas (Project no. 2018-1745). We thank the TofTools team for providing tools for mass spectrometry data analysis.





**Table A1.** Overview of experimental conditions. AS = ammonium sulfate, ABS = ammonium bisulfate, eff = effloresced seed, deli = deliquesced seed. Condensation sinks are calculated for $C_{10}H_{16}O_7$ from the dry size distribution, and listed separately for the steady states before ($SS_1$) and during ($SS_2$) seed injection. For the beginning of Experiment 7, the DMPS was malfunctioning and CS is thus not given.

| # | T (K) | RH (%) | [AP] (ppb) | [O$_3$] (ppb) | [NO$_X$] (ppb) | [NO] (ppt) | Seed | Seed state | $CS_{SS_1}$ (s$^{-1}$) | $CS_{SS_2}$ (s$^{-1}$) |
|---|-------|--------|------------|---------------|----------------|------------|------|------------|------------------------|------------------------|
| 2 | 302 | <1 | 33 | 70 | 0 | 0 | AS | eff | $0.90\times10^{-3}$ | $9.0\times10^{-3}$ |
| 3 | 302 | 44 | 22 | 79 | 0 | 0 | AS | eff | $1.0\times10^{-3}$ | $15\times10^{-3}$ |
| 4 | 303 | 40 | 22 | 80 | 0 | 0 | ABS | eff | $2.1\times10^{-3}$ | $11\times10^{-3}$ |
| 5 | 303 | 44 | 21 | 78 | 0 | 0 | ABS | deli | $2.4\times10^{-3}$ | $12\times10^{-3}$ |
| 6 | 302 | 47 | 22 | 79 | 0 | 0 | AS | deli | $2.1\times10^{-3}$ | $14\times10^{-3}$ |
| 7 | 302 | 45 | 23 | 79 | 0 | 0 | ABS | deli | NA | $8.8\times10^{-3}$ |
| 8 | 302 | 45 | 22 | 78 | 0 | 0 | AS | eff | $1.6\times10^{-3}$ | $10\times10^{-3}$ |
| 9 | 302 | 42 | 23 | 78 | 0 | 0 | AS | eff | $1.5\times10^{-3}$ | $12\times10^{-3}$ |
| 10 | 303 | 45 | 21 | 77 | 0 | 0 | ABS | eff | $2.2\times10^{-3}$ | $8.5\times10^{-3}$ |
| 11 | 301 | 47 | 35 | 75 | 0 | 0 | AS | eff | $1.5\times10^{-3}$ | $5.6\times10^{-3}$ |
| 12 | 300 | <1 | 35 | 82 | 0 | 0 | AS | eff | $2.4\times10^{-3}$ | $8.6\times10^{-3}$ |
| 13 | 301 | <1 | 34 | 75 | 0 | 0 | ABS | eff | $2.2\times10^{-3}$ | $9.3\times10^{-3}$ |
| 14 | 301 | <1 | 35 | 66 | 0 | 0 | ABS | eff | $2.2\times10^{-3}$ | $9.9\times10^{-3}$ |
| 15 | 302 | 46 | 24 | 73 | 0 | 0 | ABS | deli | $3.0\times10^{-3}$ | $7.7\times10^{-3}$ |
| 16 | 302 | <1 | 91 | 40 | 33 | 110 | AS | eff | $1.2\times10^{-3}$ | $9.5\times10^{-3}$ |
| 21 | 302 | 46 | 61 | 40 | 25 | 120 | ABS | deli | $0.73\times10^{-3}$ | $7.5\times10^{-3}$ |
| 17 | 302 | 42 | 59 | 48 | 25 | 200 | ABS | deli | $1.1\times10^{-3}$ | $7.9\times10^{-3}$ |
| 18 | 301 | 43 | 60 | 47 | 25 | 200 | AS | deli | $1.1\times10^{-3}$ | $9.1\times10^{-3}$ |
| 19 | 302 | <1 | 86 | 51 | 33 | 180 | AS | eff | $1.0\times10^{-3}$ | $9.1\times10^{-3}$ |
| 20 | 302 | <1 | 85 | 48 | 33 | 180 | ABS | eff | $1.2\times10^{-3}$ | $12\times10^{-3}$ |

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
