# Peer review of "Experimental investigation into the volatilities of highly oxygenated organic molecules (HOM)"

_Atmospheric Chemistry and Physics, 2019_

## Referee Comment (RC1) · Anonymous Referee #1 · 25 Aug 2019

This manuscript presents an experimental investigation of the volatilities of highly oxygenated organic compounds (HOM) formed in the ozonolysis of $\alpha$-pinene. The condensation behaviors of HOM upon seed injection in continuous flow chamber experiments were interpreted using box modelling to offer insights into their volatilities. The authors found that HOM dimers, along with the majority of HOM monomers are of low or extremely low volatility, while a small fraction of HOM monomers are semi-volatile. The authors further developed a parameterization for assessing the volatilities of HOM using their molecular composition, and compared the results with those derived by existing volatility parameterizations. This manuscript provides valuable information on the volatility of HOM derived from $\alpha$-pinene oxidation and also a methodology for the

determination of volatility of organic species, especially for thermally labile species. I recommend the publication of this manuscript in ACP after the authors address several important issues as detailed below.

Major comments:

P16, Sect. 3.5. As shown in Fig. 6, the possible reactive or solubility-driven uptake of oxidation products resulted in a difference in the condensation behavior of some of HOM (both CHO compounds and organonitrates) between experiments with efflo-resced and deliquesced seeds. How big was this difference? To what extent this difference affected the determination of the fraction remaining of gas-phase HOM upon seed addition and hence their volatilities? A quantitative evaluation of the effect of particle-phase processes of HOM should be included in the revised manuscript.

In addition, since the HOM monomers and dimers may decompose after condensation, producing more volatile products that may partition back to the gas phase, I am curious if the authors observed any oxidation products showing an increase in their gas-phase concentrations upon seed addition.

P18, Sect. 3.6.1. In my understanding, the oxidation products that were possibly affected by the particle-phase processes represented a source of uncertainty in volatility determination and should be excluded from the model. However, these species were actually included in the model. The authors should explain this.

P21, L11. The authors stated that they couldn't exclude the role of particle-phase processes in artificially lowering HOM volatility estimated using their parameterization. Have the authors tried to develop a parameterization excluding HOM species affected by particle-phase processes from the model and compare the results with existing volatility estimates?

Minor comments:

P7, L16. I suggest the authors provide details as to how the wall loss lifetime of ELVOC

was estimated from their condensation behavior upon seed addition.

P8, L12-14. Since the lifetime of gas-phase ELVOC depends on the condensation sink (CS). The authors should specify the value of CS leading to a 60

P12, L20. Replace "as well as" by "and".

P15, Fig. 5 and P17, Fig. 6. A logistic fit to the data in these figures may help to demonstrate the trend more obviously and also help to locate the transition mass between the high- and low-volatility limits.

P16, L6. Delete "in the differences".

P19, L4. Delete "the".

P24, Table A1. Why the numbering of experiments starts with 2? And the numbering of exps. 17-21 is not in numerical order.

---

## Referee Comment (RC2) · Anonymous Referee #2 · 29 Aug 2019

Highly oxygenated organic molecules (HOM) play an important role in new particle formation, early growth and constitute a large fraction of secondary organic aerosol. To assess their role in these processes better, their volatilities should be known. However, their structures could so far not be identified and there are no easily accessible surrogate compounds. In this paper a method is presented to determine the volatility of HOMs. Ozonolysis of alpha-pinene was performed in a simulation chamber until a steady state concentration of HOMs was reached. After injection of a seed aerosol a new steady state concentration of HOMs was obtained. From the difference of the concentration of the two steady-states the volatility of HOMs could be derived. Using the chemical composition of the HOMs a relation between their volatility and their car-

bon, hydrogen, nitrogen and oxygen numbers was derived. It was found that volatility does decrease less with addition of an oxygen atom than predicted by other parameterizations reported in literature. Furthermore, the experiments were well simulated with the ADCHAM model. The results presented in this study are of high interest and well suited for ACP. The method used here is well suited and the experiments and data analysis were well done. The paper is also well written and I recommend publication of this manuscript. There are a few points I would like the authors to clarify or add some more information and I also suggest some more additions to put the results presented here into the perspective of other works.

Major comments: The main assumption for the analysis is that the source terms stay constant upon seed addition. The authors note that RO2 radicals decrease by less than 20%. Now, the main reaction path to HOM for NOx-free conditions is by RO2+ RO2, since the RO2 concentration is much higher than HO2. Thus, the HOM formation rate could decrease considerably (40% max), if the total RO2 concentration would decrease by 20%. Could you comment on this.

Page 16, line 33ff: Do you see difference for both types of seeds at 40% RH compared to 1% RH? It is difficult to see the differences from Figure 6. Would it be better visible using a scatter plot between the two RH systems? The condensation sink is calculated from the dry seed diameter. At 40% RH the particles are larger. How much would this influence the fraction remaining?

Figure 7: The observed values seem to be systematically lower than the modeled ones at lower fraction remaining. It looks like there is still a sigmoidal dependence. How was the weighting of the signals done? Does it strongly influence the fit?

Page 20, line 1ff: I suggest to also show in this discussion an example for a known compound of higher volatility, e.g. pinic acid, pinonic acid. I have the feeling that this relationship only holds for compounds with multiple peroxy groups.

Table 1: It is difficult to grasp all the information in the table. I suggest to present it

also in a figure O:C versus log C*, as it is often done in literature. This could make the dependencies better visible. Furthermore, could you also compare your volatilities with other measurements, e.g. from FIGAERO (D'Ambro et al., Earth & Space Chemistry 2018, 2, 1058-1067; Schobesberger et al., ACP 18, 14757-14785, 2018). Although these measurements are from the particle phase they still cover similar types of compounds.

Page 22 line 11; Figure A2: The authors suggest to use their new parameterization in future studies. In the simulation of Figure A2, their model uses the SIMPOL parameterization. I wonder, how their new parameterization would affect the simulation? Does it give similar results?

Minor comments: Page 1, line 18: Replace hyrdoxy by hydroxyl. This also occurs several times later in the text.

Page 4, line 20: Did you also correct for hygroscopic growth as in Dal Maso et al.?

Page 12, line 10; Here you say that you excluded compounds with SD-to-signal ratio higher than 4. How can you evaluate a signal at such high noise? In Figure 5 it says compounds with a signal to noise ratio above 4 have been excluded and on page 18, line 13 it states compounds with a signal to noise ratio below 4 have been excluded. I assume only the last statement is correct.

Figure 2: Color of shading is different from what is stated in the Figure caption.

Figure 3: You report here experiment seven. How can you model this without knowledge of CS during SS1?

Figure 5: Is the scaling linear?

---

## Author Comment (AC1) · 26 Oct 2019

We thank the two anonymous reviewers for the insightful and constructive comments and questions regarding our manuscript. We have addressed the reviewers' comments point-by-point below:

[Figure]

**1   Anonymous Referee 1**

This manuscript presents an experimental investigation of the volatilities of highly oxygenated organic compounds (HOM) formed in the ozonolysis of $\alpha$-pinene. The condensation behaviors of HOM upon seed injection in continuous flow chamber experiments were interpreted using box modelling to offer insights into their volatilities. The authors found that HOM dimers, along with the majority of HOM monomers are of low or extremely low volatility, while a small fraction of HOM monomers are semi-volatile. The authors further developed a parameterization for assessing the volatilities of HOM using their molecular composition, and compared the results with those derived by existing volatility parameterizations. This manuscript provides valuable information on the volatility of HOM derived from $\alpha$-pinene oxidation and also a methodology for the determination of volatility of organic species, especially for thermally labile species. I recommend the publication of this manuscript in ACP after the authors address several important issues as detailed below.

**Major comments:**

**1.1   Referee comment 1**

P16, Sect. 3.5. As shown in Fig. 6, the possible reactive or solubility-driven uptake of oxidation products resulted in a difference in the condensation behavior of some of HOM (both CHO compounds and organonitrates) between experiments with efflo-resced and deliquesced seeds. How big was this difference? To what extent this difference affected the determination of the fraction remaining of gas-phase HOM upon seed addition and hence their volatilities? A quantitative evaluation of the effect of particle-phase processes of HOM should be included in the revised manuscript.

[Figure]

**1.1.1 Author response**

The referee correctly identifies one of the limitations of the study, that is, the possibility of uptake of HOM driven by heterogeneous or particle phase processes, and not by volatility.

1. As for the more quantitative assessment of the difference between the effloresced and deliquesced cases, we have added a new subplot to Figure 6, showing a scatter plot of the fractions remaining between these two conditions. This should aid in assessing the magnitude of the difference and was something that referee 2 also requested.

2. As for the effect this has on the determination of the fraction remaining, and thus the assessment of the volatility of HOM: from the difference between the efflo-resced and deliquesced conditions, we know that there is an enhanced uptake of HOM in the deliquesced case, presumably caused by particle phase processes. Even the calculation of the condensation sink is more complicated in the humid case: however, we expect this effect to be minor (see response to reviewer 2, comment 2). However, we do not know the extent, if any, that particle phase processes play in the effloresced case. Therefore, the difference between the two conditions does not directly tell us anything about the potential particle phase processes in the effloresced case, just that there are some in the deliquesced case. Because of this, we use the effloresced, non-acidic seed conditions for the parametrization of the volatility. The possibility of particle phase processes affecting this condition as well is an inherent limitation of the study. By choosing the experiments with non-acidic, effloresced seed, we try to minimize the effect of this as best we can.
**1.1.2 Changes to the manuscript**

We have added a new subplot, detailing the difference between the effloresced and deliquesced cases (see response to comment 2 from reviewer 2). The possibility of particle phase processes affecting the results is acknowledged in the section "2.4.4. Effect of heterogeneous chemistry on the gas phase", including its last sentence (already present in the original manuscript): "Still, we cannot fully exclude the effect of particle phase processes on the response of HOM to seed addition.", and in other parts of the manuscript.

**1.2 Referee comment 2**

In addition, since the HOM monomers and dimers may decompose after condensation, producing more volatile products that may partition back to the gas phase, I am curious if the authors observed any oxidation products showing an increase in their gas-phase concentrations upon seed addition.

**1.2.1 Author response**

We do indeed observe the increase of the concentrations of some compounds upon seed addition: these are visible with fractions remaining above one in Figures 5 and 6. This type of behaviour is only observed for a small number of compounds. However, we may not be sensitive to others that exhibit similar behaviour. We have added additional discussion on this in the manuscript. Also, we have excluded the compounds with FR over 1.1 from the statistical model.

[Figure]

**1.2.2 Changes to the manuscript**

We have added the following text when discussing the Figure 5:
"*Indeed, we observe some compounds with the values for fraction remaining above one, mainly at lower masses (Fig. 5): this implies an increased source term upon seed addition. Presumably these compounds are formed in the particle phase and are of high enough volatility to evaporate back to gas phase, as discussed in Sect. 2.4.4. However, the number of these compounds is small, and they lie mainly outside the HOM masses. So, while there are indications of some particle phase processes taking place, we expect that the volatility of a compound is dominant in determining their behaviour for the vast majority of the detected compounds.*"

**1.3 Referee comment 3**

P18, Sect. 3.6.1. In my understanding, the oxidation products that were possibly affected by the particle-phase processes represented a source of uncertainty in volatility determination and should be excluded from the model. However, these species were actually included in the model. The authors should explain this.

P21, L11. The authors stated that they couldn't exclude the role of particle-phase processes in artificially lowering HOM volatility estimated using their parameterization. Have the authors tried to develop a parameterization excluding HOM species affected by particle-phase processes from the model and compare the results with existing volatility estimates?

**1.3.1 Author response**

As noted in response to referee comment 1, we cannot readily distinguish those compounds that are affected by particle phase processes in the effloresced case. We only

know that there are indications that some compounds are affected in the deliquesced case. Excluding those compounds would not be justified, as we do not know the extent to which, if any, they are affected in the effloresced case. However, as noted in the reply to the previous comment, there were some compounds whose concentration increased during seed addition: this is a clear indication of particle phase processes affecting those compounds. We have now chosen to exclude any compounds with a fraction remaining above 1.1 (meaning a 10 % increase upon seed addition) from the statistical model, as these are clearly influenced by particle phase processes. However, there was only one such compound, so this plays no large role in determining the model fit.

**1.3.2   Changes to the manuscript**

We have excluded compounds with FR over 1.1 from the model, and added the following sentence in section 3.6.1: "*In addition, any compounds with a FR value over 1.1 (meaning a 10 % increase upon seed addition) were excluded due to the influence of particle phase processes on them.*"

We have also added the following sentences to section 3.6.1: "*Finally, the possibility that the uptake of some compounds to particles is not driven by their volatility, but rather some particle phase processes (as noted in Sect. 2.4.4) would affect the modelling as well. This would lead to artificially low volatility estimates. However, as noted above, we have tried to minimize this effect.*"

**Minor comments:**

**1.4 Referee comment 4**

P7, L16. I suggest the authors provide details as to how the wall loss lifetime of ELVOC was estimated from their condensation behavior upon seed addition.

**1.4.1 Author response**

ELVOC concentration is determined by their source strength and their lifetime in the gas phase, as noted in the manuscript. Assuming a constant source, any changes in the concentration are caused by changes in the lifetime. During a seed injection, the ELVOC concentration drops to a fraction of its original value: this drop is determined by the ratio of the gas phase lifetimes during and before seed injection. As the main determinants of the lifetime are the condensation sink and the wall loss, and we know the condensation sink, we can then use the known condensation sink and drop in ELVOC concentrations to estimate the wall loss. We validated this approach by comparing to the results of Ehn et al. (2014, 10.1038/nature13032), where the wall loss was directly measured. They used UV lights to produce high concentrations of OH radicals, which then reacted with a-pinene. This led to HOM formation, which could be instantaneously shut off by turning off the light. The wall loss lifetime was then calculated from the rapid decay of the gas phase HOM. There, the wall loss estimated from ELVOC drop during seed injection agrees strikingly well with the one estimated from UV switch off experiments. We have added details on the wall loss determination in the manuscript.

**1.4.2 Changes to the manuscript**

We reformatted the paragraph discussing the gas phase lifetime of ELVOC. The main changed text is: "*A typical condensation sink caused by particles formed in the chamber in the absence of inorganic seed was* $2 \times 10^{-3}$ *$s^{-1}$ (Table A1), corresponding to*

*a lifetime of 500 seconds with respect to the loss to particle surfaces. When adding seed particles, the typical condensation sink was $10 \times 10^{-3}\ s^{-1}$. This corresponds to a lifetime of only 100 seconds with respect to the condensation to particle surfaces. Thus, the losses to condensation on aerosol particles are, to a first approximation, an order of magnitude faster than either the chemical sink or flush out. We do not have a direct measurement of the wall loss lifetime in the chamber. However, we can estimate it from the behaviour of ELVOC upon seed addition. Without any wall loss, the sink term of ELVOC would increase roughly fivefold, reflecting directly on the gas phase concentrations. However, the observed decrease in concentrations is smaller. A wall loss lifetime of 400 s explains the observed decrease in ELVOC well: this number is consistent across experiments. This was also a free parameter in the ADCHAM model, which yielded identical results.*"

**1.5 Referee comment 5**

P8, L12-14. Since the lifetime of gas-phase ELVOC depends on the condensation sink (CS). The authors should specify the value of CS leading to a 60

**1.5.1 Author response**

We have added the numbers.

**1.5.2 Changes to the manuscript**

The text now reads: "*Upon a typical seed injection experiment, the condensation sink increases from around $2 \times 10^{-3}\ s^{-1}$ to around $10 \times 10^{-3}\ s^{-1}$, and condensation onto aerosol particles becomes the main sink of ELVOC. This leads to the decrease of the gas phase lifetime of ELVOC by around 60 % (Fig. 1).*"

**1.6 Referee comment 6**

P12, L20. Replace "as well as" by "and".

**1.6.1 Author response**

Replaced

**1.7 Referee comment 7**

P15, Fig. 5 and P17, Fig. 6. A logistic fit to the data in these figures may help to demonstrate the trend more obviously and also help to locate the transition mass between the high- and low-volatility limits.

**1.7.1 Author response**

This is something we considered when preparing the manuscript, but eventually decided against. This was for a couple of reasons. First, we present a logistic fit on the composition of the HOM later in the text. This is a better fit, as, like stated in the text, it is not really the mass of a compound that determines its volatility, but rather its chemical makeup. Presenting two logistic fits could be confusing to the reader, who might pick up on the relationship between the mass and the fraction remaining and thus volatility, instead of the more proper one between the molecular formula and the volatility. Also, for Fig. 6 a fit for the non-nitrates and organic nitrates separately would make more sense: this would be a hybrid between the fit on mass only, and the one on elemental composition. We feel it is better to stick to the fit on elemental composition. Further, as more of a subjective opinion, we feel that the trend is rather obvious already, and a fit would not add much value to the figure, while increasing the complexity unnecessarily.

Therefore, we decided to keep the figures without the fits.

**1.8 Referee comment 8**

P16, L6. Delete "in the differences".

**1.8.1 Author response**

Deleted

**1.9 Referee comment 9**

P19, L4. Delete "the".

**1.9.1 Author response**

Deleted

**1.10 Referee comment 10**

P24, Table A1. Why the numbering of experiments starts with 2? And the numbering of exps. 17-21 is not in numerical order.

**1.10.1 Author response**

This was a tentative numbering, used in the analysis phase. This included a failed, non-listed experiment 1, as well as the addition of experiment 21 later, when we realized

that there actually was good gas-phase data from that experiment as well, contrary to our initial impression. For clarity, we have changed the numbering to start from 1 and proceed in numerical order through the experiments, in chronological order.

1.10.2   Changes to the manuscript

We have changed the numbering, both in the table and in the text.

 

**2   Anonymous Referee 2**

Highly oxygenated organic molecules (HOM) play an important role in new particle formation, early growth and constitute a large fraction of secondary organic aerosol. To assess their role in these processes better, their volatilities should be known. However, their structures could so far not be identified and there are no easily accessible surrogate compounds. In this paper a method is presented to determine the volatility of HOMs. Ozonolysis of alpha-pinene was performed in a simulation chamber until a steady state concentration of HOMs was reached. After injection of a seed aerosol a new steady state concentration of HOMs was obtained. From the difference of the concentration of the two steady-states the volatility of HOMs could be derived. Using the chemical composition of the HOMs a relation between their volatility and their carbon, hydrogen, nitrogen and oxygen numbers was derived. It was found that volatility does decrease less with addition of an oxygen atom than predicted by other parameterizations reported in literature. Furthermore, the experiments were well simulated with the ADCHAM model. The results presented in this study are of high interest and well suited for ACP. The method used here is well suited and the experiments and data analysis were well done. The paper is also well written

and I recommend publication of this manuscript. There are a few points I would like the authors to clarify or add some more information and I also suggest some more additions to put the results presented here into the perspective of other works.

**Major comments:**

2.1   Referee comment 1

The main assumption for the analysis is that the source terms stay constant upon seed addition. The authors note that RO2 radicals decrease by less than 20 %. Now, the main reaction path to HOM for NOx-free conditions is by RO2+ RO2, since the RO2 concentration is much higher than HO2. Thus, the HOM formation rate could decrease considerably (40% max), if the total RO2 concentration would decrease by 20 %. Could you comment on this.

2.1.1   Author response

The 20 % decrease was an upper end estimate based on an RO2 lifetime of 10 seconds, with a high increase in CS during seed injection. This was meant more as an upper limit, but we now realize that it seems unnecessarily high. For a more typical increase of CS, the drop in RO2 lifetime would be less than 10 %. In addition, in the ADCHAM model the RO2 lifetimes are closer to 5 seconds, which would further decrease the drop.

Also, for most of the HOM monomers at least, the formation is probably through a reaction of a highly oxidized RO2 (HOM-RO2) with an early, much less oxidized RO2. What happens to these upon contact with seed particles is to our knowledge unknown. However, given that RO2 radicals in aqueous solution typically terminate through bimolecular reactions (https://doi.org/10.1016/S0273-1223(97)00003-6), we expect that the uptake of those RO2 to particles is not especially fast. If this is the case, and the less oxidized RO2 are not affected by seed addition, the drop in HOM source term for most compounds would come only from the drop in HOM-RO2 radicals. We performed some simulations using ADCHAM with enhanced uptake of the less oxidized RO2 to particles, but this deteriorated the fit between the model and the observations considerably. Thus, it seems that the less oxidized RO2 indeed are not much affected by the seed addition.

For the highly oxidized HOM dimers, which require two HOM-RO2 to form, the decrease in source term would be quadratic to the HOM-RO2 decrease, as noted. However, we observe no clear trend of increasing drop in concentrations with increasing oxygen content of the dimers. Therefore, we assume this effect to be minor, but still worth mentioning as a limitation in the manuscript. We also added an example from Garmash et al. (2019, acpd, 10.5194/acp-2019-582): in an analogous situation, they observe a larger than predicted drop of HOM upon seed addition: this is explained by multi-generation OH oxidation, where both the precursor and the HOM itself drop upon seed addition, resulting in a larger HOM loss from the gas phase.

**2.1.2   Changes to the manuscript**

We re-formatted the paragraph discussing RO2 loss to seed particles. We added the following sentences: "*Upon contact with seed particles or chamber walls, similarly to ELVOC, the highly oxidized RO$_2$ can be expected to be lost from the gas phase. However, this only becomes an important sink for them if their chemical lifetime is long enough to allow for non-negligible condensation. If this is the case, seed addition may influence their concentration.*"

And changed the sentence with the 20 % drop to: "*During a typical seed injection, the gas phase lifetime of RO$_2$, and thus their concentration, are expected to drop by less*

*than 10 %*"

And added the following subsection: "**A note on multi-generation oxidation** *We have so far considered oxidation products originating directly from VOC oxidation, through short-lived RO$_2$ intermediates. In the case of HOM from $\alpha$-pinene, this is a good approximation (Bianchi et al., 2019). In contrast, in some systems, oxidation products may undergo repeated oxidation by e.g. hydroxyl radicals, leading to production of more oxidized products. This is observed in the case of aromatics Garmash et al. (2019). In this case, both the HOM formed in the repeated oxidation, and the precursor, itself an oxidation product, may condense on seed particles. Garmash et al. (2019) observed some compounds dropping more than expected upon seed addition, and explained this in terms of multi-generation oxidation. This is a clear example where the decrease of a gas phase compound upon seed addition does not only depend on its volatility, but the volatility of its precursors as well. However, in the case of $\alpha$-pinene the vast majority of HOM form directly from the oxidation of $\alpha$-pinene, and thus this effect should be minor (Bianchi et al., 2019).*"

**2.2   Referee comment 2**

Page 16, line 33ff: Do you see difference for both types of seeds at 40% RH compared to 1% RH? It is difficult to see the differences from Figure 6. Would it be better visible using a scatter plot between the two RH systems? The condensation sink is calculated from the dry seed diameter. At 40% RH the particles are larger. How much would this influence the fraction remaining?

**2.2.1   Author response**

Yes, the difference is similar in AS dry vs. wet and in ABS dry vs. wet. Further, there is no big difference between AS wet and ABS wet, like there is no big difference between

AS and ABS dry.

We agree that judging the differences between experiments by eye is challenging and subjective: as suggested, we have added scatter plots to aid in the comparisons of experiments. We added two scatter plots: the first details the (non)difference between the non-nitrate monomers in the no-NOx and NOx experiments (so between the first and the second sigmoid plot), as Figure 6b. The second details the difference between the dry vs. humid cases (the two sigmoid plots in the figure 6): this is the new figure 6d.

It is true that the condensation sink is calculated for the dry particles, and this is an underestimation in the humid case. The exact values of the condensation affect the fraction remaining for nonvolatile species, with higher condensation sink giving lower fraction remaining. With increasing CS, the position of the "lower arm" of the sigmoid curve will shift downwards: however, the general behavior should not be affected. Also, in the further analyses for explaining the fraction remaining, we used only the dry experiments. Therefore, we feel that trying to account for the effect of RH on the condensation sink would bring unnecessary complexity to the analyses, without contributing to the results or their interpretation.

**2.2.2   Changes to the manuscript**

We have added two new subplots to the Figure 6: the new figure, along with its updated caption, is also shown at the end of the document.

In addition, we have added a few lines comparing the findings for the AS vs ABS seed to our earlier study: "*However, in the same set of experiments, Riva et al. (2019) found a large SOA enhancement on dry ABS seed particles. The lack of difference in the gas-phase HOM concentrations indicates that the increase in SOA did not come from enhanced HOM uptake, as measured by the $NO_3^-$-CI-APi-TOF. Indeed, Riva et*

*al. (2019) observe a marked decrease of more volatile gas-phase oxidation products, including pinonaldehyde, upon ABS seed addition.*"

**2.3 Referee comment 3**

Figure 7: The observed values seem to be systematically lower than the modeled ones at lower fraction remaining. It looks like there is still a sigmoidal dependence. How was the weighting of the signals done? Does it strongly influence the fit?

**2.3.1 Author response**

The fitting of the model was done using the Matlab function fitglm, and the weighting of the observations using the option "Weight". In essence, this fits a generalized linear model, giving more weight to the compounds with a high signal in the estimation. This is good for two reasons: their values for fraction remaining are less uncertain, and in this way the model better describes the majority of the HOM observed. We further investigated the behaviour of the fitted vs. observed values for fraction remaining and have updated Fig. 7. Most of the compounds for which the observed values were lower than the modelled ones have a carbon number lower than 10. In contrast, most C10 compounds are fitted well, and probably play a large role in determining the model fit. This can indicate that the dependence of the fraction remaining (FR) on the oxygen number is not the same for all carbon numbers (e.g. that the FR for C8 compounds would drop more steeply upon oxygen addition than it does for C10 compounds). This could, to some extent, be described in the model by including an appropriate interaction term. However, this would complicate the model quite a bit. We feel that the current, simple model describes the data well enough not to justify a more complex model. We have added discussion on this in the manuscript. We have also changed the signal-to-noise criterion to be stricter for the compounds to be included in the model.

**2.3.2  Changes to the manuscript**

We have updated figure 7: the updated version is shown below. The sentence "*Larger deviances from the 1:1 line are mainly explained by carbon numbers lower than ten.*" was added to the caption.

We have also added the following discussion in section 3.6.1: "*Out of the monomers, the abundant $C_{10}$ compounds are fitted the best, appearing close to the 1:1 line. The compounds deviating from this line mainly have a smaller carbon number: $C_9$ compounds seem especially problematic for the model. This could be an indication that the dependence of the volatility on e.g. the oxygen number is different for compounds with fewer than 10 carbon atoms. Both organic nitrates and non-nitrates seem to be fitted equally well.*"

**2.4  Referee comment 4**

Page 20, line 1ff: I suggest to also show in this discussion an example for a known compound of higher volatility, e.g. pinic acid, pinonic acid. I have the feeling that this relationship only holds for compounds with multiple peroxy groups.

**2.4.1  Author response**

We agree that this relationship is probably specific for autoxidation products. Older parametrizations, specifically fitted for compounds of higher volatility, may work better for non-autoxidation systems. We have now added an explicit mention of this in the manuscript. Also, we have added the examples of pinic and pinonic acid in the discussion.

**2.4.2 Changes to the manuscript**

We have added the following sentences to the section in question: "*Also, the relationship presented in Eq. (11) may not hold for oxidation systems other than the one presented here, or indeed compounds formed in $\alpha$-pinene oxidation, but not through autoxidation. Examples of products formed in the latter way include pinic ($C_9H_{14}O_4$) and pinonic acids ($C_{10}H_{16}O_3$). Equation (11) gives them saturation concentrations 15 $\mu g\ m^{-3}$ and 27 $\mu g\ m^{-3}$, respectively, compared to literature values in the range of a couple of $\mu g\ m^{-3}$ for pinic and from less than one to up to hundreds of $\mu g\ m^{-3}$ for pinonic acid (Bilde and Pandis, 2001; Salo et al., 2010; Donahue et al., 2012). Especially for pinonic acid the spread in literature values is very large. Given that HOM are chemically quite different from the two acids, the agreement is surprisingly good. Still, any generalizations should be drawn with caution.*"

**2.5 Referee comment 5**

Table 1: It is difficult to grasp all the information in the table. I suggest to present it also in a figure O:C versus log C*, as it is often done in literature. This could make the dependencies better visible. Furthermore, could you also compare your volatilities with other measurements, e.g. from FIGAERO (D'Ambro et al., Earth Space Chemistry 2018, 2, 1058-1067; Schobesberger et al., ACP 18, 14757-14785, 2018). Although these measurements are from the particle phase they still cover similar types of compounds.

**2.5.1 Author response**

We agree that the table was difficult to grasp and have replaced it with a figure containing the same information.

[Figure]

As noted, in Schobesberger et al. and D'Ambro at al. the volatility measurements are from the particle phase. The method is based on thermal desorption, and in all cases presented the authors needed thermal decomposition of oligomers, in addition to the evaporation of free monomers, to explain the thermograms. Thus, the thermo/evapograms were always explained as a combination of at least two volatilities. In addition, while the compounds they investigate are similar to HOM, they mainly have less oxygen. Due to these reasons, we have decided not to include comparison to those measurements.

**2.5.2 Changes to the manuscript**

We have replaced the table with a plot, see the end of the document. The caption remains almost the same.

**2.6 Referee comment 6**

Page 22 line 11; Figure A2: The authors suggest to use their new parameterization in future studies. In the simulation of Figure A2, their model uses the SIMPOL parameterization. I wonder, how their new parameterization would affect the simulation? Does it give similar results?

**2.6.1 Author response**

We realize that the wording for the suggestion was too strong, and have reworded.

We tested to use our parametrization for the volatilities of HOM in the ADCHAM model, but this did not change the results significantly. In the ADCHAM model, we used a nucleation parametrization from Kirkby et al. (2016, 10.1038/nature17953), that assumes

all non-nitrate HOMs to nucleate. Hence, the parametrized nucleation rate does not depend on the volatilities of HOM. Further, in the presence of seed, most HOM will condense anyway. Therefore, further investigations are required to assess the effect of the exact volatility parametrization on particle formation.

**2.6.2 Changes to the manuscript**

The sentence suggesting to use our parametrization now reads: "*Future studies should evaluate the effect of the exact volatility parametrization used on new particle formation from HOM.*"

**Minor comments:**

**2.7 Referee comment 7**

Page 1, line 18: Replace hyrdoxy by hydroxyl. This also occurs several times later in the text.

**2.7.1 Author response**

Corrected the typos

**2.8 Referee comment 8**

Page 4, line 20: Did you also correct for hygroscopic growth as in Dal Maso et al.?

**2.8.1 Author response**

No, we used the dry condensation sink directly. See also response to referee 2, comment 2. Dal Maso et al. have a parametrization to typical Hyytiälä hygroscopic growth, which would most probably be wrong in our case. Also, as noted in the previous reply, the exact values of the condensation sink are not too important for the conclusions.

**2.8.2 Changes to the manuscript**

We have added the following sentences to the part in question: "*We did not correct for hygroscopic growth of the particles, so in humid cases the calculated condensation sink is an underestimate: however, this should not have any notable effect on the conclusions.*"

**2.9 Referee comment 9**

Page 12, line 10; Here you say that you excluded compounds with SD-to-signal ratio higher than 4. How can you evaluate a signal at such high noise? In Figure 5 it says compounds with a signal to noise ratio above 4 have been excluded and on page 18, line 13 it states compounds with a signal to noise ratio below 4 have been excluded. I assume only the last statement is correct.

**2.9.1 Author response**

Yes, the wording was all messed up, and only the last statement was correct. Now we have corrected these, and in addition changed the S/N criterion for the binomial model to be more strict (exclude compounds below ten).

**2.10 Referee comment 10**

Figure 2: Color of shading is different from what is stated in the Figure caption.

**2.10.1 Author response**

Caption fixed

**2.11 Referee comment 11**

Figure 3: You report here experiment seven. How can you model this without knowledge of CS during SS1?

**2.11.1 Author response**

The experiment was mislabeled by accident: in reality, this was experiment nine (where the CS is known). And with the new numbering scheme of the experiments (see referee 1, comment 10), it becomes experiment eight.

**2.11.2 Changes to the manuscript**

Corrected the numbering

**2.12 Referee comment 12**

Figure 5: Is the scaling linear?

**2.12.1 Author response**

Yes, surface area scales linearly with signal. We have now added an explicit mention of this in the captions of figures 5 - 7.

[Figure]

[Figure]

**Fig. 1.**

[Figure]

Fig. 2.

[Figure]

[Figure]

**Fig. 3.**